# Quantifying the photocurrent fluctuation in quantum materials by shot noise

Longjun Xiang [1], Hao Jin[1] & Jian Wang [1,2,3] ✉

The DC photocurrent can detect the topology and geometry of quantum materials without inversion symmetry. Herein, we propose that the DC shot noise (DSN), as the fluctuation of photocurrent operator, can also be a diagnostic of quantum materials. Particularly, we develop the quantum theory for DSNs in gapped systems and identify the shift and injection DSNs by dividing the second-order photocurrent operator into off-diagonal and diagonal contributions, respectively. Remarkably, we find that the DSNs can not be forbidden by inversion symmetry, while the constraint from time-reversal symmetry depends on the polarization of light. Furthermore, we show that the DSNs also encode the geometrical information of Bloch electrons, such as the Berry curvature and the quantum metric. Finally, guided by symmetry, we apply our theory to evaluate the DSNs in monolayer GeS and bilayer $MoS_2$ with and without inversion symmetry and find that the DSNs can be larger in centrosymmetric phase.

It is well known that the materials without inversion ($\mathcal{P}$) symmetry under light illumination can feature the bulk photovoltaic effect (BPVE)[1–6], which refers to the DC photocurrent generation in a single-phase material, such as in bulk ferroelectric perovskite oxides[7–10] and in two-dimensional (2D) piezoelectric materials[11–14]. The intrinsic physical origin of BPVE usually attributes to the shift and injection current mechanisms[6], which are closely related to the quantum geometry of Bloch electrons[15–25]. Recently, theoretical advances unveil that the $\mathcal{PT}$-invariant materials[26–29], such as the 2D antiferromagnetic insulators $CrI_3$[26] and $MnBi_2Te_4$[27], in which the $\mathcal{P}$-symmetry and the time-reversal ($\mathcal{T}$) symmetry are broken individually, can also exhibit a BPVE due to the magnetic injection current mechanism[26–28].

Beyond its importance for BPVE, the photocurrent, which is the ultimate result of photoexcitation (a typical multiphysics process), also carries a large amount of information about light-matter interaction and hence can be viewed as a diagnostic for the multiphysics process occurred in quantum materials[30]. For example, the electrons in **K** and **K′** valleys of gapped Dirac materials, such as bilayer graphene[31–33] and monolayer transition metal dichalcogenides[34,35], can be selectively excited by the left-hand or right-hand circularly polarized light due to the opposite Berry curvature in **K** and **K′** and thereby the photocurrent can be used to detect the quantum geometry of

$\mathcal{P}$-broken Dirac materials. In addition, the circular injection current in $\mathcal{P}$-broken Weyl semimetals can be exploited to measure the topological charge of Weyl cone[36]. However, as dictated by its $\mathcal{P}$-odd characteristic, the shift and injection DC photocurrent can not be employed to diagnose the quantum materials with $\mathcal{P}$-symmetry, such as the centrosymmetric topological insulators and Dirac materials[37].

On the other hand, the quantum fluctuation of photocurrent, which usually behaves as the noise of photocurrent, remains rarely explored[38], although the noise is ubiquitous in the process of none-quilibrium photoexcitation and transport. The noise is often deemed to be detrimental to the detected signal and needs to be optimized, but it can also convey information about the investigated system[39]. For example, it has been well-established that the shot noise (SN) can probe the quantum statistics of the quasi-particles and measure their effective charge in mesoscopic systems[39–43]. In addition, the SN has been used to reveal the topological phase transition of 2D semi-Dirac materials[44] and to probe the nonlocal hot-electron energy dissipation[45].

Compared to the current, the SN (as the current correlation) usually features a different symmetry requirement and hence may offer a complementary probe to detect the responses of quantum materials, particularly with $\mathcal{P}$-symmetry. To that purpose, we develop

[1]College of Physics and Optoelectronic Engineering, Shenzhen University, Shenzhen, China. [2]Department of Physics, University of Hong Kong, Hong Kong, China. [3]Department of Physics, The University of Science and Technology of China, Hefei, China. ✉e-mail: jianwang@hku.hk

**Table 1 | The constraint from $\mathcal{P}$-symmetry, $\mathcal{T}$-symmetry, and $\mathcal{PT}$-symmetry for the DC shot noise (DSN) excited by linearly or circularly polarized light**

|  | $\sigma_L$ | $\sigma_C$ | $\eta_L$ | $\eta_C$ |
|---|---|---|---|---|
| $\mathcal{P}$ | ✓ | ✓ | ✓ | ✓ |
| $\mathcal{T}$ | ✗ | ✓ | ✓ | ✗ |
| $\mathcal{PT}$ | ✗ | ✓ | ✓ | ✗ |

Here ✓(✗) means that the DSN susceptibility tensors which contain $\sigma_{L/C}$ and $\eta_{L/C}$ defined in Eq. (1) are even (odd) under symmetry operation. Here the even (odd) tensor is allowed (forbidden) by the corresponding symmetry. Note that all DSNs feature the $\mathcal{P}$-even characteristic and the DSNs in $\mathcal{T}$-invariant systems feature the same behavior with that in $\mathcal{PT}$-invariant systems.

the quantum theory of DC SN (DSN) in this work. Here the DSN means the zero frequency component of SN due to the fluctuation of photocurrent. In particular, we identify that at the second order of optical electric field, the DSN also contains the shift and injection contributions, which can be formally expressed as

$$S^{(2)} = \delta(\Omega_1)\left(\sigma_{L/C} + t_0\eta_{L/C}\right)\delta_{L/C}, \tag{1}$$

where $\Omega_1$ is the response frequency, $t_0$ the effective illumination time, $\sigma$ ($\eta$) the susceptibility tensor of the shift (injection) DSN. In addition, $\delta_L \equiv |\mathbf{E}|^2$ and $\delta_C \equiv |\mathbf{E}\times\mathbf{E}^*|$ stand for the linearly polarized light (LPL) and the circularly polarized light (CPL)[4], respectively. As indicated by the subscripts $L$ and $C$ of $\sigma$ and $\eta$, we note that $\sigma$ and $\eta$ can be excited with LPL or CPL, which depends on the $\mathcal{T}$-symmetry of the investigated systems, as summarized in Table 1. Remarkably, we find that both $\sigma$ and $\eta$ are $\mathcal{P}$-even tensors, which means that the DSNs can survive in $\mathcal{P}$-invariant systems, in sharp contrast with their DC photocurrent counterparts. Moreover, we reveal that the DSNs are also characterized by the band geometrical quantities, such as the local Berry curvature and the local quantum metric, similar to the DC photocurrent. Finally, we illustrate our formulation by investigating the DSNs in monolayer GeS and bilayer MoS$_2$ with and without $\mathcal{P}$-symmetry using first-principles calculations.

## Results

### The quantum theory for DSNs

Within independent particle approximation[16,17], the second-quantization photocurrent operator along $a$ direction at the $i$th order of optical electric field $\mathbf{E}(t)$ (Specifically, we consider the monochromatic optical field $E^b(t) = E^b(\omega_\beta)e^{-i\omega_\beta t} + c.c.$ with $\omega_\beta$ the driving frequency of light field and $c.c.$ the complex conjugate of the first term.) is defined as ($e = \hbar = 1$)

$$\hat{j}^{a,(i)}(t) \equiv \sum_{nm}\int_k \hat{\rho}^{(i)}_{mn}(t)v^a_{nm} = \sum_{nm}\int_k J^{a,(i)}_{mn}(t)a^\dagger_m a_n, \tag{2}$$

where $a^\dagger_m/a_n$ is the creation/annihilation operator, $v^a_{nm}$ the velocity matrix element, $\int_k = \frac{1}{V}\int d\mathbf{k}/(2\pi)^d$ (Here $V$ and $d$ stand for the system volume and dimension, respectively). In addition, $\hat{\rho}^{(i)}_{mn}(t) \propto |\mathbf{E}|^i$ is the second-quantization density matrix element operator and $J^{a,(i)}_{mn}(t) \propto |\mathbf{E}|^i$ the matrix element for the second-quantization photocurrent operator, see Supplementary Note 1 subsections (1.1) and (1.2), respectively.

Note that Eq. (2) is obtained by "quantizing" the statistical information (or electron occupation information) of current expectation value $J^a \equiv \text{Tr}[\hat{\rho}\hat{v}^a] = \int_k \rho_{mn}v^a_{nm}$, where $\hat{\rho}$ and $\hat{v}^a$ stand for the first-quantization density matrix operator and current operator, respectively. Particularly, since the electron occupation information is fully encoded in the density matrix element $\rho_{mn}$, at the zeroth order of $\mathbf{E}(t)$, we immediately obtain $\hat{\rho}^{(0)}_{mn} = a^\dagger_m a_n$ due to $\rho^{(0)}_{mn} = \delta(0)\delta_{nm}f_m = \langle a^\dagger_m a_n\rangle_s^{16,17}$, where $\langle\cdots\rangle_s$ stands for the quantum statistical average and $f_m$ is the equilibrium Fermi distribution function. Furthermore, by requiring $\langle\hat{\rho}^{(i)}_{mn}(t)\rangle_s = \rho^{(i)}_{mn}(t)^{16}$, $\hat{\rho}^{(i)}_{mn}(t)$ with $i\geq1$ can be obtained by

iteratively solving the Liouville equation from $\hat{\rho}^{(0)}_{mn} = a^\dagger_m a_n$, where the time dependence of $\hat{\rho}^{(i)}_{mn}$ arises from $\mathbf{E}(t)$ whereas $a^\dagger_m/a_n$ does not evolve with time. In addition, we wish to remark that $\hat{j}^{a,(i)}$ have considered the quantum average but retained the statistical informaton in operator form and hence $\text{Tr}[\cdots]$, which contains both the quantum average and quantum statistical average, can not be used to calculate the average of $\hat{j}^{a,(i)}$ and $\hat{j}^{a,(i)}\hat{j}^{b,(j)}$.

We emphasize that Eq. (2), together with the quantum statistical average $\langle\cdots\rangle_s$, is designed to evaluate the photocurrent and photocurrent correlation on the equal footing, by following the noise formulation developed in mesoscopic conductors[39]. Particularly, for the second-order photocurrent operator $\hat{j}^{a,(2)}$, by dividing it into off-diagonal and diagonal contributions[46] in term of the interband and intraband contributions of $v^a_{nm}$, namely, by writing $\hat{j}^{a,(2)}(t) = \hat{j}^{a,(2)}_O(t) + \hat{j}^{a,(2)}_D(t)$ with $\hat{j}^{a,(2)}_O(t) \equiv \sum_{nm}^{m\neq n}\int_k\hat{\rho}^{(2)}_{mn}(t)v^a_{nm}$ and $\hat{j}^{a,(2)}_D(t) \equiv \sum_n\int_k\hat{\rho}^{(2)}_{nn}(t)v^a_{nn}$, we find that the quantum statistical average of $\hat{j}^{a,(2)}_O$ and $\hat{j}^{a,(2)}_D$ give the familiar DC shift and injection photocurrent [see Supplementary Note 1 subsection (1.3)], respectively, and thereby we define $\hat{j}^{a,(2)}_O$ and $\hat{j}^{a,(2)}_D$ as the shift and injection photocurrent operator, respectively. Interestingly, we find that $\hat{j}^{a,(2)}_O$ and $\hat{j}^{a,(2)}_D$ will further give DSNs (dubbed the shift and injection, respectively) at the second order of $\mathbf{E}(t)$, as illustrated below.

With Eq. (2), by evaluating the photocurrent operator correlation function defined by[39] $S^{ab}(t,t') \equiv \frac{1}{2}\langle\Delta\hat{j}^a(t)\Delta\hat{j}^b(t') + \Delta\hat{j}^b(t')\Delta\hat{j}^a(t)\rangle_s$, where $\Delta\hat{j}^a(t) \equiv \hat{j}^a(t) - \langle\hat{j}^a(t)\rangle_s$, the SN spectrum is given by [see Supplementary Note 2 subsection (2.1)]

$$S^{a,(i+j)}(t,t') = \frac{1}{2}\sum_{nm}\int_k J^{a,(i)}_{nm}(t)J^{a,(j)}_{mn}(t')f^2_{nm}, \tag{3}$$

where $S^{a,(i+j)} \equiv S^{aa,(i+j)}$ (we focus on the autocorrelation of photocurrent operator) and $f_{nm} \equiv f_n - f_m$. Note that $S^{a,(i+j)}(t,t')$ is a function of two independent time variables and generally stands for an AC SN. However, by adopting a Wigner transformation[39], we obtain a new correlation function $S^{a,(i+j)}(t_1,t_0)$, where $t_1 = t - t'$ and $t_0 = (t+t')/2$ represent the short and long time scale, respectively. Next, by taking the time average over $t_0$, we can pick up the DC component of $S^{a,(i+j)}(t_1,t_0)$ for $t_0$ because we are interested in the noise spectrum on a time scale long compared to $1/\omega$, where $\omega$ is the driving frequency of $\mathbf{E}(t)$. To be specific, we have

$$S^{a,(i+j)}(t_1) = \frac{1}{T}\int_0^T dt_0 S^{a,(i+j)}(t,t')\Big|_{t'=t_0-t_1/2}^{t=t_0+t_1/2}, \tag{4}$$

where $T \equiv 2\pi/\omega$. Moreover, by performing a Fourier transform for $S^{a,(i+j)}(t_1)$, we obtain the SN spectrum $S^{a,(i+j)}(\Omega_1)$, where $\Omega_1$ is the response frequency for $t_1$. As expected, at the second order of $\mathbf{E}(t)$, we extract a DSN $S^{a,(2)}(\Omega_1) = \delta(\Omega_1)S^{a,(2)}$, where $S^{a,(2)}$ is only contributed by the equal-time correlation between $\hat{j}^{a,(0)}$ and $\hat{j}^{a,(2)}$ [see Supplementary Note 2 subsection (2.2)]. We wish to mention that the strategy to extract the DC component from a general double-time correlation function $S^{a,(i+j)}(t,t')$ is the same as that adopted in mesoscopic conductors[39]. Finally, we note that $S^{a,(2)}$ contains the shift and injection contributions, which, respectively, further contains the $\mathcal{T}$-even and $\mathcal{T}$-odd components. For simplicity we will only display their $\mathcal{T}$-even expressions that survive in $\mathcal{T}$-invariant systems, while their $\mathcal{T}$-odd counterparts can be found in Supplementary Note 2 subsections (2.3) and (2.4).

Particularly, for shift DSN, by defining $S^{a,(2)}_{sht} \equiv \sum_{\omega_\beta = \pm\omega}\sigma^{abc}$ $(0;\omega_\beta, -\omega_\beta)E^b(\omega_\beta)E^c(-\omega_\beta)$, the $\mathcal{T}$-even shift DSN susceptibility tensor

is given by [see Supplementary Note 2 subsection (2.3)]

$$\sigma_C^{abc} = \frac{\pi}{4} \sum_{mn} \int_k f_{nm}^2 \left( W_{mn}^{abc} - W_{mn}^{acb} \right) \delta(\omega - \omega_{mn}), \quad (5)$$

where $\hbar\omega_{mn} = \hbar(\omega_m - \omega_n)$ is the energy difference between bands $m$ and $n$. Here $W_{mn}^{abc} \equiv i(v_{mn;b}^a r_{nm;a}^c - v_{nm;b}^a r_{mn;a}^c)$ with $O_{nm;b}^a = \partial_b O_{nm}^a - i(\mathcal{A}_n^b - \mathcal{A}_m^b)O_{nm}^a$, where $O = v, r$ and $\mathcal{A}_n^b/r_{nm}^a$ is the intraband/interband Berry connection.

As indicated by the subscript $C$ in $\sigma^{abc}$, we note that the $\mathcal{T}$-even shift DSN can only be excited by CPL due to $\sigma_C^{abc} = -\sigma^{acb}$[22]. Similar to the shift photocurrent, we find that the shift DSN is closely related to the band geometrical quantities. To see that, we note that $v_{mn;b}^a = i\omega_{mn}r_{mn;b}^a + i\Delta_{mn}^b r_{mn}^a$[17], where $\Delta_{mn}^a = v_m^a - v_n^a$ is the group velocity difference. Then by substituting its second term into $W_{mn}^{abc}$, we find that $W_{mn}^{abc} = \Delta_{nm}^b(g_{nm}^{ac}\partial_a \ln |r_{nm}^c| + \Omega_{nm}^{ac}R_{nm}^{a,c})$, where $g_{nm}^{ac} = r_{nm}^a r_{mn}^c + r_{nm}^c r_{mn}^a$ is the local quantum metric[29], $\Omega_{nm}^{ac} \equiv i(r_{nm}^a r_{mn}^c - r_{nm}^c r_{mn}^a)$ the local Berry curvature[27], and $R_{nm}^{a,c} = -\partial_a \phi_{nm}^c + \mathcal{A}_n^a - \mathcal{A}_m^a$ with $\phi_{nm}^c = r_{nm}^c/|r_{nm}^c|$ the shift vector[16]. Interestingly, in sharp contrast with the shift photocurrent, we find that $\sigma_C^{abc}$ is also related to the group velocity difference $\Delta_{nm}^a$, which usually appears in the injection photocurrent. Finally, we emphasize that $\sigma_C^{abc}$ is a rank-4 tensor since $\sigma^{abc} \equiv \sigma^{aabc}$, where the first index $a$ is responsible for the direction of autocorrelated photocurrent. The same convention will be applied to the injection DSN discussed below.

Similarly, for injection DSN, by defining $\partial_{t_0} S_{inj}^{a,(2)} \equiv \sum_{\omega_\beta = \pm\omega} \eta^{abc} (0; \omega_\beta, -\omega_\beta)E^b(\omega_\beta)E^c(-\omega_\beta)$, we find that the $\mathcal{T}$-even component of $\eta^{abc}$ is given by [see Supplementary Note 2 subsection (2.4)]

$$\eta_L^{abc} = \frac{\pi}{4} \sum_{nm} \int_k f_{nm}^2 \Delta_{mn}^a (I_{mn}^{abc} + I_{mn}^{acb}) \delta(\omega - \omega_{mn}), \quad (6)$$

where $I_{mn}^{abc} = i(v_{mn;b}^a r_{nm}^c - v_{nm;b}^a r_{mn}^c)$. Note that the $\mathcal{T}$-even injection DSN can only be excited by LPL due to $\eta^{abc} = \eta^{acb}$[22]. Importantly, also by substituting the second term of $v_{mn;b}^a = i\omega_{mn}r_{mn;b}^a + i\Delta_{mn}^b r_{mn}^a$ into $I_{mn}^{abc}$, we hvae $I_{mn}^{abc} = \omega_{nm}(g_{nm}^{ca}\partial_b \ln |r_{nm}^a| + \Omega_{nm}^{ca}R_{nm}^{b,a}) + \Delta_{nm}^b g_{nm}^{ac}$, which means that the injection DSN is not only related to the local Berry curvature and the local quantum metric like the injection photocurrent, but also related to the shift vector, which usually appears in the shift photocurrent.

In summary, Eqs. (5)-(6) constitute the quantum theory for the DSN at the second order of $\mathbf{E}(t)$ in $\mathcal{T}$-invariant systems. As expected, we find that Eqs. (5)-(6) are gauge-invariant under $U(1)$ gauge transformation. Importantly, by using the sum rules of $r_{mn;b}^a$ and $v_{mn;b}^a$[13], we find that Eqs. (5)-(6) can be employed to investigate the quantum fluctuation of photocurrent operator in realistic materials by combining them with first-principles calculations. Note that in Eqs. ((5)-(6)), $e = \hbar = 1$ has been adopted. By dimension analysis, a universal factor $e^4/\hbar^2$ must be recovered for first-principles calculations. To guide the calculation, we next discuss the symmetry constraints for the DSN susceptibility tensors in $\mathcal{T}$-invariant systems.

## The symmetry for DSNs

The symmetry plays a pivotal role in the discussion of the DC photocurrent[47]. For example, under $\mathcal{P}$-symmetry, we have $\mathcal{P}J^a = -J^a$ and $\mathcal{P}E^b = -E^b$, and therefore the DC photocurrent proportional to $|\mathbf{E}|^2$ vanishes in $\mathcal{P}$-invariant systems[25]. In addition, the magnetic injection (shift) photocurrent can only be generated in both $\mathcal{P}$-broken and $\mathcal{T}$-broken materials under the illumination of LPL (CPL), in which the magnetic injection (shift) photocurrent susceptibility is a $\mathcal{P}$-odd as well as $\mathcal{T}$-odd tensor[27].

In the above, we have established the quantum theory for the DSNs in $\mathcal{T}$-invariant systems, but the symmetry constraint on these tensors has not been fully discussed yet. Particularly, under $\mathcal{P}$-symmetry, we have $\mathcal{P}\Delta_{mn}^a = -\Delta_{mn}^a$, $\mathcal{P}v_{mn}^a = -v_{mn}^a$, $\mathcal{P}v_{mn;b}^a = v_{mn;b}^a$,

$\mathcal{P}r_{mn}^a = -r_{mn}^a$, and $\mathcal{P}r_{mn;b}^a = r_{mn;b}^a$. And thereby the DSN susceptibility tensors given in Eqs. ((5)-(6)) feature the $\mathcal{P}$-even characteristic, which is also true for their $\mathcal{T}$-odd counterparts, as summarized in Table 1. Note that the $\mathcal{P}$-even property can also be obtained from the general response relation, where both $J$ and $E$ appear twice. Notably, the $\mathcal{P}$-even characteristic dictates that all the DSNs are immune to $\mathcal{P}$-symmetry, in sharp contrast with the $\mathcal{P}$-odd shift or injection photocurrent, as expected. Intuitively, as a general feature of photocurrent, the $\mathcal{P}$-symmetry must be broken either by crystal structure or by external perturbation to guarantee that the left-going and right-going photocurrents cannot cancel with each other. However, for SN, this cancellation mechanism is lifted since the correlation of current is nonzero even when the current is zero, as exemplified by the notable Nyquist-Johnson noise in mesoscopic conductors[39].

Similar to the DC photocurrent, the constraint on DSNs from $\mathcal{T}$-symmetry is tricky because we need to take into account the polarization of light at the same time. For example, under LPL (CPL), we find that the shift photocurrent susceptibility is a $\mathcal{T}$-even ($\mathcal{T}$-odd) tensor while the injection photocurrent susceptibility is a $\mathcal{T}$-odd ($\mathcal{T}$-even) tensor. Therefore, in $\mathcal{T}$-invariant but $\mathcal{P}$-broken systems, one can only detect the shift (injection) photocurrent by illuminating LPL (CPL). Eqs. ((5)-(6)) give the $\mathcal{T}$-even DSNs, which can be easily checked by using $\mathcal{T}\Delta_{mn}^a = -\Delta_{mn}^a$, $\mathcal{T}v_{mn}^a = -v_{mn}^a$, $\mathcal{T}v_{mn;b}^a = v_{mn;b}^a$, $\mathcal{T}r_{mn}^a = r_{mn}^a$, and $\mathcal{T}r_{mn;b}^a = -r_{mn;b}^a$, as summarized in Table 1, where their $\mathcal{T}$-odd counterparts are also listed as a comparison. In addition, dictated by their $\mathcal{P}$-even characteristic, we find that Eqs. (5)-(6) also feature the $\mathcal{P}\mathcal{T}$-even property and hence can be applied to investigate the DSN in $\mathcal{P}\mathcal{T}$-invariant materials, sharply different from the DC photocurrent.

Besides the $\mathcal{P}$, $\mathcal{T}$, and $\mathcal{P}\mathcal{T}$ symmetries, to consider the constraint from the point group (PG) symmetry, such as the rotation and mirror symmetries, one should resort to the Neumann's principle[48], which determines the non-vanishing tensor elements under PG symmetry operation. Particularly, for the rank-4 SN susceptibility tensor $\lambda^{abcd}$ in $\mathcal{T}$-invariant systems, where $\lambda$ stands for $\sigma$ and $\eta$, the constraint imposed by PG symmetry operation $R$ can be expressed as:

$$\lambda^{abcd} = R_{aa'}R_{bb'}R_{cc'}R_{dd'}\lambda^{a'b'c'd'}, \quad (7)$$

where $R_{\alpha\alpha'}$ is the matrix element of $R$. For example, if the system respects mirror symmetry $\mathcal{M}_x$ with $\mathcal{M}_x x \to -x$, one can immediately realize that $\lambda^{yyxz}$ is forbidden in terms of Eq. (7). Alternatively, one can also use the Bilbao Crystallographic Server[49] to identify the non-vanishing tensor element for all PGs just by defining a suitable Jahn notation. Particularly, the Jahn notation for the $\mathcal{T}$-even rank-4 shift (injection) DSN $\sigma_C^{abc}$ ($\eta_L^{abc}$), which is anti-symmetric (symmetric) about the last two indices, can be expressed as $VV\{V^2\}$ ($VV[V^2]$), where $V$ represents the polar vector and $\{\cdots\}$ ($[\cdots]$) indicates the anti-symmetric (symmetric) permutation symmetry, as listed in Table 2. As a comparison, we also list the Jahn notations for shift and injection photocurrent susceptibility tensors in $\mathcal{T}$-invariant systems. Based on symmetry analysis, we are ready to explore the DSNs in realistic materials.

## The monolayer GeS

As the first example, we explore the DSNs in single-layer monochalcogenide GeS, which has been extensively studied[19,50,51] and displayed a large BPVE and spontaneous polarization in its ferroelectric phase with PG $mm2$, as shown in the upper panel of (Fig. 1a). The PG $mm2$ doesn't contain the $\mathcal{P}$-symmetry and hence both the shift and injection photocurrents are allowed. However, besides the ferroelectric phase, GeS may stay in a paraelectric phase with PG $mmm$, as shown in the lower panel of (Fig. 1a), which respects $\mathcal{P}$-symmetry and hence can not generate a DC photocurrent. In (Fig. 1b), we display the band structures for GeS with $mm2$ and $mmm$ symmetries and we find that its band structure goes through a large modification from the

**Table 2 | The Jahn notations for shift ($\sigma_C$) and injection ($\eta_L$) DC shot noise susceptibility tensors in $\mathcal{T}$-invariant systems**

|  | $\sigma_C$ | $\eta_L$ | $\sigma_{2L}$ | $\eta_{2C}$ |
|---|---|---|---|---|
| Jahn notations | $VV\{V^2\}$ | $VV[V^2]$ | $V[V^2]$ | $V\{V^2\}$ |

For comparison, we also list the $\mathcal{T}$-even shift and injection photocurrent susceptibility tensors, which are represented by $\sigma_{2L}$ and $\eta_{2C}$, respectively. For brevity, we have suppressed their superscripts.

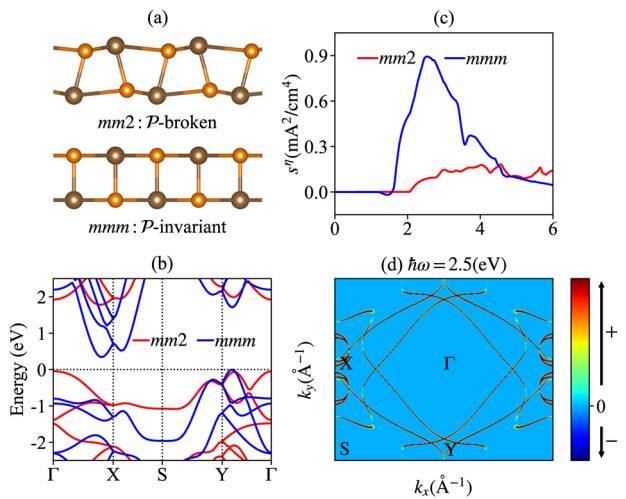

**Fig. 1 | DC shot noise (DSN) for monolayer GeS. a** The side views of monolayer GeS with point group (PG) *mm2* and *mmm*, where *mm2/mmm* breaks/respects the $\mathcal{P}$-symmetry. **b** The band structures for GeS with different PGs. The Fermi level indicated by the horizontal dashed line is placed on the top of the valence band. **c** The injection DSN $s^\eta = t_0 \eta_L^{xyy} E^2$ for GeS with different PGs, where $V = 1\,\mathrm{cm}^3$, $t_0 = 10^{-14}\,\mathrm{s}$ and $E = 10^7/\sqrt{2}\mathrm{V/m}$[57]. **d** The **k**-resolved integrands for the $\eta_L^{xyy}[\hbar\omega = 2.5(\mathrm{eV})]$ only for PG *mmm*.

ferroelectric phase to the paraelectric phase. However, since the paraelectric phase can not generate a DC photocurrent response thus the band structure evolution can not be detected just by measuring the photocurrent. In addition, we find that the shift DSN susceptibility tensor $\sigma_C^{abc}$ is also forbidden by mirror symmetry $\mathcal{M}_x$ or $\mathcal{M}_y$ in both *mm2* and *mmm* if $b \neq c$ or by the antisymmetric permutation symmetry if $b = c$. We wish to mention that $\sigma_C^{abc}$ is a subset of the rank-4 tensor $\sigma_C^{abcd}$ and the full table for $\sigma_C^{abcd}$ is not forbidden by symmetry according to the Jahn notation listed in Table 2.

However, the injection DSN, such as $\eta_L^{xyy}$, is allowed by both *mmm* and *mm2* phases of GeS due to its symmetric permutation symmetry about the last two indices, as shown in (Fig. 1c). From (Fig. 1c), we find that the injection DSN (denoted as $s^\eta = t_0 \eta_L^{xyy} E^2$) in *mmm* plays a dominant role compared with that in *mm2*. Particularly, we find that $s^\eta$ reaches a peak with photon energy $\hbar\omega = 2.5$ (eV). Furthermore, by plotting the **k**-resolved integrand for $s^\eta$ in (Fig. 1d), we identify that the main peak is contributed by the optical transitions around X and Y points in the Brillouin zone. The different DSN responses for GeS in its $\mathcal{P}$-invariant and $\mathcal{P}$-broken phases may offer a tool to probe the band structure evolution from the ferroelectric phase to the paraelectric phase. Finally, we remark that there are other independent symmetry-allowed elements for the injection DSN of GeS, see Supplementary Note 3 and Supplementary Fig. 1.

**The bilayer MoS$_2$**
As the second example, we explore the DSNs in bilayer MoS$_2$ within 2H and 3R phases, as shown in (Fig. 2a). The bilayer 2H-MoS$_2$ respects $\mathcal{P}$ symmetry and hence both the shift and injection photocurrents are forbidden in this system. However, bilayer MoS$_2$ could possess a

$\mathcal{P}$-broken phase by modifying the stack configuration of constituent monolayers. Very recently, it is shown that bilayer MoS$_2$ with 3R stacking pattern could exhibit an out-of-plane electric polarization, which is also known as the sliding ferroelectricity[52–55]. In (Fig. 2b), we show the band structures for bilayer MoS$_2$ within 3R and 2H phases. Different from monolayer GeS discussed before, we find that the band structures of bilayer MoS$_2$ are almost unchanged with these two different stacking patterns.

Also different from the monolayer GeS, we find that both shift and injection DSN (denoted as $s^\sigma = \sigma_C^{xxz} E^2$ and $s^\eta = t_0 \eta_L^{xxx} E^2$, respectively) exist for both $\mathcal{P}$-broken 3R and $\mathcal{P}$-invariant 2H bilayer MoS$_2$, even when the mirror symmetry exists in 3R and 2H. In particular, in (Fig. 2c-d) we plot the shift DSN $s^\sigma$ and injection DSN $s^\eta$ for bilayer MoS$_2$ with 3R and 2H phases, respectively. Interestingly, we find that the injection DSN in bilayer MoS$_2$ features almost the same behavior in 2H and 3R phases while the shift DSN is dominant in $\mathcal{P}$-invariant phase, where the first peak at $\hbar\omega = 2.4$(eV) is contributed by the optical transitions around $\Gamma$ point, as shown in (Fig. 2e). In (Fig. 2f), we also display the **k**-resolved distribution for the peak of $\eta_L^{xxx}$ located at $\hbar\omega = 2.2$(eV). Similarly, besides $\sigma_C^{xyy}$ and $\eta_L^{xxx}$, there are other symmetry-allowed DSNs components in bilayer MoS$_2$ with or without $\mathcal{P}$-symmetry, see Supplementary Note 3 and Supplementary Fig. 2. Note that the spin-orbit coupling is ignored in above discussions for simplicity, whose influence is discussed in the Supplementary Information (see Supplementary Note 4 and Supplementary Fig. 3).

## Discussion
The DSNs discussed in this work originates from the light irradiation so that the relaxation processes (the photoexcited electrons lose their energy and then relax to the conduction band edge of gapped systems) usually play a key role, particularly to distinguish different contributions[56]. Specifically, we have two DC contributions at the second order of **E**($t$), namely the shift and injection DSNs, which arise from the off-diagonal and diagonal components of $\hat{j}^{a,(2)}$, respectively. Therefore, the corresponding relaxation processes should resemble the shift and injection DC photocurrents. In particular, the shift photocurrent and DSN, which stand for intrinsic contributions[6], are less relevant to the impurity scattering. However, the injection photocurrent and DSN usually are related to the complicated scattering processes when relax to the edge of conduction band and are governed by a relaxation time about $10^{-12}$ to $10^{-14}$ s[56]. In this work, the relaxation process for injection DSN is modeled by a constant relaxation time, similar to the DC photocurrent discussed in Refs.[26], [57], and [58]. At this stage, we wish to remark that the extrinsic SN of shift current photovoltaics discussed in Ref.[38] is different from our results, which can be seen from the following aspects: (i) their formulation is based on steady-state assumption[18] while our formulation does not assume that; (ii) their shot noise formula for shift photocurrent does not contain the key geometric quantity−shift vector, which is believed to be the physical origin of shift photocurrent.

Although our formulation is developed by following the scattering matrix theory in mesoscopic conductors[39], the results show some different features. First, we note that it is not easy to derive a relation similar to the Schottky formula[39,59] within our bulk formulation by adopting a general approximation. Second, the symmetry in our bulk formulation plays an essential role, which dictates that the SN and the photocurrent barely appear at the same time under the assigned polarization of light, whereas the symmetry usually are less important in mesoscopic transport systems. These differences may be attributed to that our bulk formulation does not include the effect of electrodes that are inevitably involved in experimental measurements, while the scattering matrix theory includes that. However, the effect of electrodes is minor in a diffusive conductor (Besides metallic conductors, here the diffusive conductor also means an insulator under light illumination, where the electrons located at valence bands are excited to

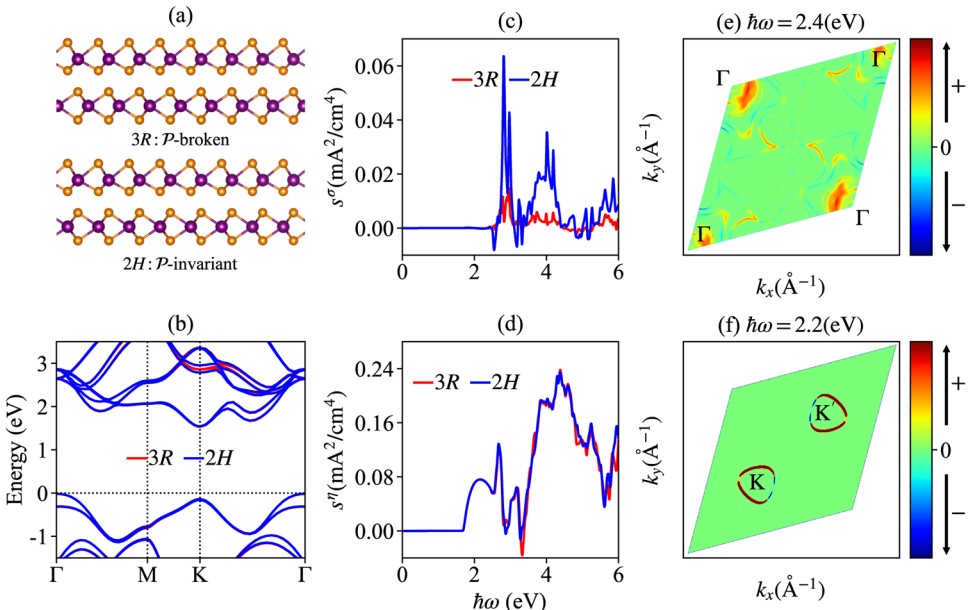

**Fig. 2 | DC shot noise (DSN) for bilayer MoS₂. a** The side views of bilayer MoS₂ with point group (PG) 3R and 2H, respectively, where 3R/2H breaks/respects $\mathcal{P}$ symmetry. **b** The band structures for bilayer MoS₂ with different PGs. The Fermi level indicated by the horizontal dashed line is placed on the top of valence band. **c, d** The shift DSN $s^\sigma = \sigma_C^{xxz}E^2$ and the injection DSN $s^\eta = t_0\eta_L^{xxx}E^2$ for bilayer MoS₂ with PG 3R and 2H, respectively, where $V = 1\,cm^3$, $t_0 = 10^{-14}(s)$, and $E = 10^7/\sqrt{2}(V/m)$[57]. **e, f** The **k**-resolved integrands for the $\sigma_C^{xxz}[\hbar\omega = 2.4(eV)]$ and $\eta_L^{xxx}[\hbar\omega = 2.1(eV)]$ only for 2H MoS₂.

the conduction bands.), as manifested by the contact resistance due to electrodes[60,61], therefore, the photocurrent DSNs based on bulk response theory can be detected in a diffusive conductor. This is indeed the case of lots of current (photocurrent) measurements[36,62–64] to verify the predictions based on the bulk formulation[20,65,66], where the symmetry plays a pivotal role to probe the quantum geometry of Bloch electrons. Furthermore, in the ballistic transport regime, the quantum geometric information may also be extracted by manipulating the symmetry[67,68].

The DC photocurrent has shown its importance in characterizing the (topological) quantum materials[36,69,70], so we expect that the DSN of photocurrent has the same importance. Although the DSN of photocurrent has not been reported experimentally, the noise spectrum of electric current in mesoscopic systems has been extensively studied experimentally for over twenty years[40,71–74]. Particularly, it has been recently realized that the current fluctuations or the shot noise induced by nonequilibrium electrons (which in our setup are driven by the optical field) will generate fluctuating electromagnetic evanescent fields on the surface of the material[45], which can be detected by using the scanning noise microscope[45,75] even without the introduction of metallic electrodes in the conventional noise measurements. Therefore, this noninvasive experimental technique can be used to verify our proposal to exclude the competing signals. Note that both the shift and injection photocurrents are forbidden in $\mathcal{P}$-invariant systems. In that case, to initiate the photocurrent correlation, an external static electric field may be applied when illuminating the insulating sample, where a "jerk" photocurrent is generated[57,76,77]. Once the photocurrent correlation is established, by gradually decreasing the static electric field, a nonzero residual DSN signal solely driven by the optical field can be expected.

Finally, we wish to remark that in materials with $\mathcal{P}$ and $\mathcal{T}$ symmetries, both the Berry curvature and the shift vector vanishes due to $\mathcal{PT}\Omega_{nm}^a = -\Omega_{nm}^a$ and $\mathcal{PT}R_{nm}^{a,c} = -R_{nm}^{a,c}$. As a consequence, the dominant geometric quantity will be the quantum metric. Recently, the experimental observation for the intrinsic nonlinear Hall effect[63,64] that driven by the quantum metric dipole has triggered much attention to explore the importance of quantum metric. Note that the intrinsic

nonlinear Hall effect can survive only in systems without $\mathcal{P}$ and $\mathcal{T}$ symmetries, while the quantum metric itself is not forbidden by them. Therefore, the formulation developed in this work exactly offers an approach to probe the quantum metric in centrosymmetric quantum materials.

In conclusion, we formulate the quantum theory to calculate the quantum fluctuation of the photocurrent operator in gapped systems. We identify the shift and injection DSNs at the second order of $\mathbf{E}(t)$ and derive their susceptibility tensor expressions that are amenable to first-principles calculation. In sharp contrast with the DC photocurrent, we find that all DSNs are allowed by the $\mathcal{P}$-symmetry due to their $\mathcal{P}$-even characteristic. In addition, we show that the DSNs also encode the information of band geometrical quantities, such as the local Berry curvature, the local quantum metric, and the shift vector. Finally, guided by symmetry, we combine our theory with first-principles calculation to estimate the DSNs in monolayer GeS and bilayer MoS₂ with and without $\mathcal{P}$-symmetry. And we find that the DSNs in $\mathcal{P}$-invariant phase can be larger than that in $\mathcal{P}$-broken phase. Our work shows that the quantum fluctuation of the photocurrent operator offers a complementary probe to characterize the quantum materials, particularly with $\mathcal{P}$-symmetry.

## Methods

First-principles calculations are performed by using the Vienna ab initio simulation package (VASP)[78]. The generalized gradient approximation (GGA) in the form of Perdew-Burke-Ernzerhof (PBE) is used to describe the exchange-correlation[79,80]. We choose 500 eV for the cutoff energy and a $k$-grid of $18 \times 18 \times 1$ for the first Brillouin zone integration. To avoid the spurious interaction, we employ at least 20 Å vacuum space along the perpendicular direction. All atoms in the supercell are fully relaxed based on the conjugate gradient algorithm, and the convergence criteria is 0.01 eV/Å for the force and $10^{-8}$ eV for the energy, respectively. A damped van der Waals (vdW) correction based on the Grimme's scheme is also incorporated to better describe the nonbonding interaction[81,82]. The maximally localized Wannier functions are then employed to construct the tight-binding model via wannier90 code, in which Mo-$d$, Ge-4$p$, and S-3$p$ orbitals are taken into

account[83,84]. The tight-binding Hamiltonian is utilized to calculate the DSNs according to Eqs. ((5)-(6)). To deal with the rapid variation of the Berry curvature, the Brillouin zone integration is carried out using a dense $k$-mesh with $600 \times 600 \times 1$, which gives well-convergent results. The 3D-like coefficients of DSN are obtained by assuming an active single-layer with a thickness of $L_a$:

$$SN_{3D} = \frac{L_{slab}}{L_a} SN_{2D}, \tag{8}$$

where $SN_{2D}$ is the calculated DSN, and $L_{slab}$ is the thickness of the supercell[84].

## Data availability
The data generated and analyzed during this study are available from the corresponding author upon request.

## Code availability
All code used to generate the plotted data is available from the corresponding author upon request.

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

## Acknowledgements

J.W. acknowledges support from the Natural Science Foundation of China (Grant No.12034014).

## Author contributions

J.W. conceived the project. L.J.X. and J.W. developed the theory and performed the symmetry analysis. H.J. and L.J.X performed the first-principles calculations. L.J.X., H.J., and J.W. wrote the paper. J.W. supervised the project. All authors analyzed the data and contributed to the discussions of the results.

## Competing interests

The authors declare no competing interests.
