## [Peer Review File · Nature Communications]

Quantifying the photocurrent fluctuation in quantum materials by shot noiseEditorial Note: Parts of this Peer Review File have been redacted as indicated to remove third-party material where no permission to publish could be obtained.

REVIEWER COMMENTS

Reviewer #1 (Remarks to the Author):

The authors study shot noise of photocurrent fluctuation that arises from photoexcitation of electrons in insulators under light irradiation. They formulate shot noise of photocurrent and discuss its connections to geometrical quantities including quantum metric and Berry curvature. They apply their formula of shot noise to GeS and MoS₂ with ab initio calculations, discussing the behavior of the shot noise with and without P symmetry.

The shot noise of photocurrent fluctuations and its relationship to quantum geometry is very interesting. The ab initio calculation of shot noise and their prediction of different behavior of shot noise with and without P symmetry are attracting. However, I have serious concerns about their formulation of shot noise under photoexcitation as I describe below.

1. The authors formulate the shot noise based on the length gauge formula of Aversa and Sipe (ref. 16).

It can be used to evaluate current expectation value, but it should not be used to compute current correlation function $\langle J(t)J(0) \rangle$. This can be seen from the form of the current operator in Eq. 2 that is defined as a product of current matrix element v_{mn} and the density matrix ρ . The evaluation of an expectation value of a physical quantity O involves the density matrix only once as $\text{Tr}[O \rho]$, thus usage of Eq. 2 is okay for the current expectation value $\text{Tr}[J \rho]$. However, if Eq. 2 is used for evaluating the correlation function, the density matrix appears twice like $\text{Tr}[J \rho J \rho]$ which does not make sense. A suitable correction to the convention is necessary. In doing so, the authors need to clarify the validity of the present formulation.

2. In Eq. (1) in the supplement, it seems that the authors effectively use the solution of Liouville equation for ρ as an expression for time evolution of $a^\dagger a$ that obeys Heisenberg equation. These two equations differ by sign and maybe it is related to the unusual convention of the current operator in Eq. (2). Specifically, J is defined with $J_{nm} a^\dagger_m a_n$ while it is usually defined as $J_{nm} a^\dagger_n a_m$. This is quite confusing, and this notation should be corrected.

3. The shot noise S defined above Eq. (3) is a correlation function of the current at the same time. However, the dc shot noise (or $\omega=0$ component of the noise power) should be defined as the t integral of current correlation function at a time difference of t , $\langle J(t)J(0) \rangle$. I think this is most crucial issue of the present manuscript. S in Eq.3 is equal time correlation function of J and corresponds to ω integral of noise power $S(\omega)$, which is not measured in experiments usually. The above point is clear by looking at the unit of their shot noise from the ab initio calculation which is $\text{Å}^2/m$ while the shot noise has a unit of $\text{Å}^2 \text{ s}$, which is clear from the conventional Shot noise in metallic conductors $S=2e I$ with the current I . So, my main point is: can the authors compute $S(\omega=0)$ with their formulation?

4. As a related issue, the present formulation of current operator in Eq.(2) could be used to compute equal time expectation value, but perhaps not applicable to compute the expectation value at different

times which is necessary to evaluate $\langle J(t)J(0) \rangle$ and gives the main contribution to the shot noise. As a side note, Sipe and Shkrebtii (ref 17) is formulating photocurrent with the time evolution of the equal time expectation value, so it will be good to use this convention rather than writing like $\rho_{mn}^{\dagger}(0) = a_n^{\dagger} a_m$. Anyway, I suspect it is not straightforward to compute in this formalism.

5. I also have a concern about the present formulation of shot noise based on the current operator in the bulk. J in Eq.(2) is the uniform $q=0$ component of the current. I am not sure this is the correct choice to evaluate the shot noise. Usually the shot noise is considered in a mesoscopic situation. For example, Souza et al [PRB 78, 155303 (2008)] treats shot noise by considering current between the contact and the system (not the current in the bulk) and derives Schottky formula $S=2eI$ using the Keldysh Green's function. As a check for validity to use the bulk current operator for defining shot noise, can the authors derive the Schottky formula $S=2eI$ by considering the metals, say 1 band system with a Fermi surface? I think this is crucial because if the present formulation cannot account for the conventional shot noise, the predicted shot noise with a relation to quantum geometry can be masked by additional shot noise from presence of the contact in a realistic situation, and perhaps not accessible in experiments.

To summarize, although the possible connection of shot noise of photocurrent and quantum geometry is intriguing, having the concerns as stated above, I do not recommend the manuscript for publication in Nature Communications.

Reviewer #2 (Remarks to the Author):

Authors describe how the DC shot noise (under the irradiation of light) in materials can be expressed in terms of geometric quantities (e.g., shift vector, quantum metric, and berry curvature). By examining the current operator (and separating it into shift and injection terms) they track various types of shot noise induced by light irradiation, delineating them into injection shift, and snap noise contributions; they also provide a symmetry analysis and surprisingly find that the shot noise is not limited to inversion broken materials unlike the "putative/claimed" photocurrent counterpart; they finish their work with detailed simulations in GeS and MoS₂.

I find the paper addresses an interesting topic with potentially useful results. The results clearly should be published in some form but it is unclear if nature communications is the right venue. There are three key issues which probably need to be addressed:

1. [significance of authors results vs Ref 60] Shot noise putatively/is claimed to be associated with photocurrent and specifically the shift photocurrent is not new. PRL 121, 267401 (Ref. 60) already discusses this. While it is clear that the authors have perhaps an updated and more general formulation (e.g., multi band vs two band), it is unclear to me whether and why this technical advance is so significant. For instance, authors say in footnote 60 that they have used a shift current operator (see page 2 for definition) as opposed to the local current operator of Ref 60. What insight does this provide? Are the authors saying that Ref. 60 results are wrong and that theirs is correct? As I understand it, all the

shift current operator is (as shown on their page 2) is that they have just divided the current operator in to diagonal and off-diagonal contributions. Does this basically allow to delineate the contributions into various names (based on the diagonal/off-diagonal contributions) allowing them to name the contributions? If this is the basic difference, then why is the naming so important? Is it symmetry relations under T and PT symmetries? Authors should make this clear to exhibit their novelty better.

2. [is DSN really related to photocurrent?] One striking thing in the authors work is that DSN is P even (see Table 1). This is really unusual since (A) DC rectified photocurrents arise from P broken materials and (B) authors claim that the DSN is related to the photocurrent generation processes. So I am confused: if the noise is there even for P even materials, why is it related to photocurrent at all? In the abstract, the authors propose that DSN is the quantum fluctuation of photocurrent. But if there is no photocurrent as forbidden by P symmetry why is this an OK characterisation? Surely, just because the current operator can be divided out into on-diagonal and off-diagonal contributions doesn't mean that they are shift or injection related? Perhaps authors could clarify. My suspicion is that the authors are actually probing shot noise under light irradiation whether or not photocurrent is generated. Can it be that this fluctuation is instead probing the optical absorption process (this is what injection and shift photocurrents ultimately stem from)? Note that optical absorption also scales with E^2 .

3. [probing DSN] How do the authors envision probing DSN? Given the DC character, do the authors envision a steady state experiment? If so, it would seem that relaxation should play a major role; a clearer explanation of this would have been nice (beyond the very short paragraph in the beginning of the discussion section). For instance, what happens when the carriers relax from the initial photo excitation energy to the band edge? Wouldn't the relaxation process play a role in the noise spectrum?

Reply to the reviewer #1:

Reviewer's comments: The authors study shot noise of photocurrent fluctuation that arises from photoexcitation of electrons in insulators under light irradiation. They formulate shot noise of photocurrent and discuss its connections to geometrical quantities including quantum metric and Berry curvature. They apply their formula of shot noise to GeS and MoS2 with ab initio calculations, discussing the behavior of the shot noise with and without P symmetry.

Authors' reply: We thank the reviewer for his/her careful reading and the positive comments on our manuscript.

Reviewer's comments: The shot noise of photocurrent fluctuations and its relationship to quantum geometry is very interesting. The ab initio calculation of shot noise and their prediction of different behavior of shot noise with and without P symmetry are attracting. However, I have serious concerns about their formulation of shot noise under photoexcitation as I describe below.

Authors' reply: We thank the reviewer for his/her positive comments about our work. Below we give a point-to-point reply to the concerns raised by the reviewer.

Concern 1: The authors formulate the shot noise based on the length gauge formula of Aversa and Sipe (ref. 16). It can be used to evaluate current expectation value, but it should not be used to compute current correlation function $\langle J(t)J(0) \rangle$. This can be seen from the form of the current operator in Eq. 2 that is defined as a product of current matrix element v_{mn} and the density matrix rho. The evaluation of an expectation value of a physical quantity O involves the density matrix only once as $\text{Tr}[O \text{ rho}]$, thus usage of Eq. 2 is okay for the current expectation value $\text{Tr}[J \text{ rho}]$. However, if Eq. 2 is used for evaluating the correlation function, the density matrix appears twice like $\text{Tr}[J \text{ rho} J \text{ rho}]$ which does not make sense. A suitable correction to the convention is necessary. In doing so, the authors need to clarify the validity of the present formulation.

Authors' reply: We thank the reviewer for this important question. Our formulation is indeed based on the DC photocurrent formalism developed by Sipe *et al.*^{1,2} under length gauge. We totally agree with the reviewer that Sipe's formulation can not be directly used to calculate the current correlation function $\langle \hat{J}^a(t) \hat{J}^b(0) \rangle$. This is because (i) $\text{Tr}[\dots]$ used in Sipe's formulation includes both the quantum average and the quantum statistical average, while the evaluation of shot noise only involves the quantum statistical average. This means that the current operator is obtained by

carrying out only the quantum average leaving the quantum statistical average of occupation operator unperformed, by following Büttiker's seminal work¹²; (ii) the current operator \hat{v} in the first-quantization form used in Sipe's formulation doesn't encode the statistical information or electron occupation information and hence can not give a suitable current correlation operator. In fact, the second-quantization current operator, defined by Eq.(2) in our original manuscript, together with **the quantum statistical average** $\langle \dots \rangle_s$ (In our original manuscript, we used $\langle \dots \rangle$ to represent the quantum statistical average, which may be confused with $\text{Tr}[\dots]$, as will be clear below.), are exactly designed to overcome these two difficulties and can be used to calculate the current expectation value and the current correlation on the equal footing. Interestingly, the second-quantization current operator is obtained by "quantizing" the statistical information or electron occupation information of current expectation value that evaluated by Sipe's formulation, as will be explained below. Note that our formulation is inspired by the shot noise theory developed in mesoscopic physics based on scattering matrix theory¹², where the very key step is also to write down the second-quantization current operator in terms of electron creation and annihilation operators since the noise is originated from the fluctuation of electron occupation number.

We first show that $\text{Tr}[\dots]$ used in Sipe's formulation contains the quantum average and quantum statistical average. In general, the expectation value of an operator \hat{O} is defined as⁴:

$$\langle \hat{O} \rangle \equiv \sum_n p_n \langle n | \hat{O} | n \rangle, \quad (1)$$

where $\langle n | \hat{O} | n \rangle$ stands for the quantum average over the quantum state $|n\rangle$, which is weighted by a probability p_n ($0 \leq p_n \leq 1$; $\sum_n p_n = 1$) that gives the expectation value of \hat{O} (This step is called quantum statistical average). By inserting the completeness relation into Eq.(1), we derive the equivalent definition of $\text{Tr}[\dots]$ to calculate the expectation value of \hat{O} :

$$\langle \hat{O} \rangle = \sum_n p_n \langle n | \hat{O} \left(\sum_m |m\rangle \langle m| \right) |n\rangle = \sum_m \langle m | \left(\sum_n |n\rangle p_n \langle n| \right) \hat{O} |m\rangle = \text{Tr}[\hat{\rho} \hat{O}], \quad (2)$$

where $\hat{\rho} \equiv \sum_n |n\rangle p_n \langle n|$ is the density matrix operator in the first-quantization form, which contains all statistical information or electron occupation information.

Based on Eq.(2), we next evaluate the current expectation value J^a (the same as Sipe's formulation), which will be further "quantized" into the second-quantization form, in that process the quantum statistical average $\langle \dots \rangle_s$ is naturally introduced. Using Eq.(2), we find ($e = \hbar = 1$):

$$J^a \equiv \langle \hat{v} \rangle = \text{Tr}[\hat{\rho} \hat{v}^a] = \sum_{nm} \int_k \langle n | \hat{\rho} | m \rangle \langle m | \hat{v}^a | n \rangle = \sum_{nm} \int_k \rho_{nm} v_{mn}^a, \quad (3)$$

where we have explicitly shown the integral over Brillouin zone, particularly $\int_k \equiv \frac{1}{V} \int d\mathbf{k}/(2\pi)^d$ with V and d the volume and dimension of system, respectively. Furthermore, by iteratively solving the Liouville equation^{1,2}

$$i\partial_t \rho_{mn}^{(i)} = \omega_{mn} \rho_{mn}^{(i)} + \left[iD_{mn}^b \rho_{mn}^{(i-1)} + \sum_l \left(r_{ml}^b \rho_{ln}^{(i-1)} + \rho_{ml}^{(i-1)} r_{ln}^b \right) \right] E^b(t), \quad (i \geq 1) \quad (4)$$

for density matrix element $\rho_{nm}^{(i)}$ at the i th ($i \geq 1$) order of the optical electric field from $\rho_{nm}^{(0)}$, we obtain the current expectation value $J^{a,(i)}$ at the same order, namely

$$J^{a,(i)} = \sum_{nm} \int_k \rho_{nm}^{(i)} v_{mn}^a. \quad (5)$$

Eq.(5) includes both quantum average and quantum statistical average, but the statistical information or the electron occupation information is fully encoded in $\rho_{nm}^{(i)}$. Importantly, since $\rho_{nm}^{(i)}$ is obtained by iteratively solving the Liouville equation from $\rho_{nm}^{(0)}$, where $\rho_{nm}^{(0)}$ can be "quantized" into $\hat{\rho}_{nm}^{(0)} = a_n^\dagger a_m$ due to²

$$\rho_{nm}^{(0)} = \delta(0) \delta_{mn} f_n = \langle a_n^\dagger a_m \rangle_s \equiv \langle \hat{\rho}_{nm}^{(0)} \rangle_s, \quad (6)$$

we find that $\rho_{nm}^{(i)}$ ($i \geq 1$) can be iteratively "quantized" into $\hat{\rho}_{nm}^{(i)}$ (≥ 1), which can also be obtained by iteratively solving the Liouville equation from $\hat{\rho}_{nm}^{(0)}$ because we require $\langle \hat{\rho}_{nm}^{(i)} \rangle_s = \rho_{nm}^{(i)}$, as done in the original Supplemental Material. In Eq.(6), $\langle \dots \rangle_s$ instead of $\langle \dots \rangle$ in our original manuscript is used to denote the quantum statistical average to distinguish $\langle \dots \rangle$ and $\text{Tr}[\dots]$ used in Eqs.(1-3). Here f_n is the equilibrium Fermi distribution function and a_n^\dagger/a_m stands for the time-independent creation/annihilation operator for Bloch state. Once the density matrix element in Eq.(5) is "quantized" into the second-quantization form, the current expectation value $J^{a,(i)}$ can be naturally "quantized" as

$$\hat{J}^{a,(i)}(t) \equiv \sum_{nm} \int_k \hat{\rho}_{nm}^{(i)}(t) v_{mn}^a \equiv \sum_{nm} \int_k J_{nm}^{a,(i)}(t) a_n^\dagger a_m, \quad (7)$$

as given by Eq.(2) in our original manuscript, where the time dependence due to the time-dependent optical electric field $E^b(t)$ and a factor $1/V$ in \int_k were missed in our original manuscript.

In addition, we note that $J_{nm}^{a,(i)} a_m^\dagger a_n$ and $J_{nm}^{a,(i)} a_n^\dagger a_m$ in fact are two equivalent definitions for the current correlation function, as will be clear below. And $\langle \dots \rangle_s$ (originally $\langle \dots \rangle$) in our formulation only stands for quantum statistical average and hence $\langle \hat{J}^a \rangle_s$ and $\langle \hat{J}^a \hat{J}^b \rangle_s$ can be calculated on

the equal footing, but $\text{Tr}[\dots]$ can no longer be used to calculate both \hat{J}^a and $\hat{J}^a \hat{J}^b$ defined in our formulation since they have already taken the quantum average into account.

Finally, we note that when evaluating the $\langle \hat{J}^a \hat{J}^b \rangle_s$ with our formulation, the second-quantization density matrix operator $\hat{\rho}_{nm}^{(i)}$ indeed appears twice, which is responsible for the correlation operator $a_n^\dagger a_m a_{n'}^\dagger a_{m'}$ and is related to $(f_n - f_m)^2$ in the final noise spectrum, given by Eq.(4) in our original manuscript. This in fact is the case of the shot noise discussed in mesoscopic conductors¹².

In summary, the second-quantization current operator Eq.(2) in our original manuscript, instead of the first-quantization current operator \hat{v} used in Sipe's formulation, can be used to calculate both the current expectation value and current correlation on the equal footing, but only through the quantum statistical average $\langle \dots \rangle_s$ instead of $\text{Tr}[\dots]$ used in Sipe's formulation.

In the revised manuscript, we have justified how the second-quantization current operator is derived based on the "quantized" density matrix. For the detailed revisions, please see the first and the second paragraphs (marked in blue) in the section of [The quantum theory for DSNs] in the revised manuscript, where the related Supplemental Materials has also been revised (marked in blue), particularly see the section (I) of the revised Supplemental Material.

Concern 2: In Eq. (1) in the supplement, it seems that the authors effectively use the solution of Liouville equation for rho as an expression for time evolution of a^\dagger that obeys Heisenberg equation. These two equations differ by sign and maybe it is related to the unusual convention of the current operator in Eq. (2). Specifically, J is defined with $J_{nm} a_m^\dagger a_n$ while it is usually defined as $J_{nm} a_n^\dagger a_m$. This is quite confusing, and this notation should be corrected.

Authors' reply — We thank the reviewer for this comment. We agree that the Liouville equation and the Heisenberg equation for the first-quantization density matrix operator $\hat{\rho}$, defined in Schrödinger picture and Heisenberg picture⁴, respectively, differ by a sign. And Eq.(1) in our original Supplemental Material is the Liouville equation for the second-quantization density matrix operator $\hat{\rho}_{nm}^{(i)}$ introduced above. Note that $\hat{\rho}_{nm}^{(i)}$ satisfies the same equation as the density matrix element $\rho_{nm}^{(i)}$ by requiring $\langle \hat{\rho}_{nm}^{(i)} \rangle_s = \rho_{nm}^{(i)}$ in our formulation. In addition, the creation and annihilation operators in $\hat{\rho}_{nm}^{(i)}$ don't evolve with time, which are obtained by "quantizing" $\rho_{nm}^{(0)}$ (keep $\rho_{nm}^{(0)}$ in its operator form before taking quantum statistical average) and further iteratively solving the Liouville' equation for $\hat{\rho}_{nm}^{(i)} (i \geq 1)$. With $\hat{\rho}_{nm}^{(i)}$, the second-quantization current operator can be

defined as:

$$\hat{J}^{a,(i)} \equiv \sum_{nm} \int_k \hat{\rho}_{nm}^{(i)}(t) v_{mn}^a \equiv \sum_{nm} \int_k J_{nm}^{a,(i)}(t) a_n^\dagger a_m, \quad (8)$$

or

$$\hat{J}^{a,(i)} \equiv \sum_{nm} \int_k \hat{\rho}_{nm}^{(i)}(t) v_{mn}^a \equiv \sum_{nm} \int_k J_{mn}^{a,(i)}(t) a_n^\dagger a_m. \quad (9)$$

In fact, these two definitions are equivalent since the noise spectrum

$$S^{a,(i+j)}(t, t') = \frac{1}{2} \sum_{nm} \int_k J_{nm}^{a,(i)}(t) J_{mn}^{a,(j)}(t') f_{nm}^2 \quad (10)$$

is not affected by interchanging n and m of $J_{nm}^{a,(i)} J_{mn}^{a,(j)}$. Eq.(10) is given by Eq.(4) in our original manuscript, but where we missed the time dependence. In our original manuscript, we have chosen the $J_{nm}^a a_m^\dagger a_n$, but $J_{nm}^a a_n^\dagger a_m$ is a more natural choice, as suggested by the reviewer. In the revised manuscript, we have used the natural one in our formulation, which is also adopted in the following reply. In addition, we have restored the time dependence of $\hat{J}^{a,(n)}$ that missed in our original manuscript, which is necessary to obtain the correct current correlation function, as also pointed out by the reviewer below.

Concern 3: The shot noise S defined above Eq. (3) is a correlation function of the current at the same time. However, the dc shot noise (or $\omega = 0$ component of the noise power) should be defined as the t integral of current correlation function at a time difference of t , $\langle J(t)J(0) \rangle$. I think this is the most crucial issue of the present manuscript. S in Eq.3 is equal time correlation function of J and corresponds to ω integral of noise power $S(\omega)$, which is not measured in experiments usually. The above point is clear by looking at the unit of their shot noise from the ab initio calculation which is A^2/m while the shot noise has a unit of $A^2 s$, which is clear from the conventional shot noise in metallic conductors $S = 2eI$ with the current I . So, my main point is: can the authors compute $S(\omega = 0)$ with their formulation?

Authors' reply: We thank the reviewer for the very insightful comments and the very important question. In our original manuscript, we indeed ignored the time dependence in the second-quantization current operator arising from the time-dependent optical electric field and taken the equal-time photocurrent correlation as the DC shot noise, as summarized by Eq.(1) in our original manuscript. As a consequence, the predicted results can not be measured in experiment and also

have a mismatched unit for the shot noise discussed here and in mesoscopic physics. By accounting for the time dependence of the second-quantization current operator and calculating the current correlation at different times, as suggested by the reviewer, namely

$$S^{a,(i+j)}(t, t') \equiv \frac{1}{2} \langle \Delta \hat{J}^{a,(i)}(t) \Delta \hat{J}^{a,(j)}(t') + \Delta \hat{J}^{a,(j)}(t') \Delta \hat{J}^{a,(i)}(t) \rangle_s, \quad (11)$$

where $\Delta \hat{J}^{a,(i)} \equiv \hat{J}^{a,(i)} - \langle \hat{J}^{a,(i)} \rangle_s$, our formulation indeed can be applied to compute the DC noise spectrum or the $\Omega_1 = 0$ component of the noise power for the time difference $t - t'$, as will be shown below. Note that ω in our formulation has been used to denote the frequency of the driving optical electric field so we use Ω_1 to denote the frequency of the noise power for the short time scale, defined by the time difference $t - t'$. As a result, the final result given by Eq.(1) in our original manuscript should be changed to:

$$S^{a,(2)}(\Omega_1) = S_1^{a,(2)}(\Omega_1) + S_2^{a,(2)}(\Omega_1) = \delta(\Omega_1) S_1^{a,(2)} + \delta(\Omega_1 - \omega) S_2^{a,(2)}. \quad (12)$$

where the first term contributed by the correlation between $\hat{J}^{a,(0)}$ and $\hat{J}^{a,(2)}(t)$ is the DC shot noise for $t - t'$ due to $\delta(\Omega_1)$, while the second term contributed by the autocorrelation between $\hat{J}^{a,(1)}(t)$ at different times is the AC shot noise for $t - t'$ due to $\delta(\Omega_1 - \omega)$. Here $\delta(\Omega_1)$ and $\delta(\Omega_1 - \omega)$ are obtained from Eq.(11) through the following three steps, according to the shot noise theory developed in mesoscopic conductors¹²:

1. Performing a Wigner transformation

$$\begin{cases} t_0 = (t + t')/2, \\ t_1 = t - t', \end{cases} \quad (13)$$

for $S^{a,(2)}(t, t')$ calculated by Eq.(11) to obtain $S^{a,(2)}(t_1, t_0)$, where t_1 and t_0 stand for the short and long time scales, respectively.

2. Since there are two time scales t_0 and t_1 in $S(t_1, t_0)$ and therefore we have two corresponding response frequencies: Ω_0 and Ω_1 . However, because we are interested in the noise spectra on a time scale long compared to $1/\omega$, where ω is the driving frequency for the optical electric field, we can take the average over t_0 , which amounts to picking up the $\Omega_0 = 0$ component or the DC component for long time scale, as was done in Ref.[12]. After taking the average over the long time scale t_0 , we obtain:

$$S^{a,(2)}(t_1) = \frac{1}{T} \int_0^T S^{a,(2)}(t_1, t_0) dt_0 \quad (T \equiv 2\pi/\omega). \quad (14)$$

3. Finally, performing a Fourier transform (FT) for the short time scale t_1 (or time difference $t - t'$) to obtain its noise power in the frequency domain:

$$S_1^{a,(2)}(t_1) \xrightarrow{\text{FT}} \delta(\Omega_1) S_1^{a,(2)}, \quad (15)$$

$$S_2^{a,(2)}(t_1) \xrightarrow{\text{FT}} \delta(\Omega_1 - \omega) S_1^{a,(2)}, \quad (16)$$

which contains both the DC and AC components for the short time scale or time difference.

Before going through these three steps, $S_1^{a,(2)}$ and $S_2^{a,(2)}$ in Eq.(12) are exactly the equal-time correlation calculated in our original manuscript, but they can no longer be directly added up since they belong to different response frequency channels for the short time scale. In addition, as pointed out by the reviewer, the unit of the equal-time correlations $S_1^{a,(2)}$ and $S_2^{a,(2)}$ is $\text{A}^2 \cdot \text{m}^{-1}$ (Note that we missed a factor $1/V$ in \int_k in our original manuscript), which is mismatched with the unit $\text{A}^2 \cdot \text{s}$ for shot noise in mesoscopic physics. In the revised manuscript, the shot noise spectrum at the second order of the optical electric field in our formulation is given by

$$S^{a,(2)}(\Omega_1) = \delta(\Omega_1) S_1^{a,(2)} + \delta(\Omega_1 - \omega) S_2^{a,(2)}, \quad (17)$$

which has the unit of $(\text{A} \cdot \text{m}^{-2})^2 \cdot \text{s}$ in 3D space by restoring the missed factor $1/V$ in \int_k and by considering $\delta(\Omega_1)$ due to the time dependence of correlation function, in parallel with the unit $\text{A}^2 \cdot \text{s}$ in mesoscopic situation. This is because we in fact are discussing the fluctuation of photocurrent density with unit $\text{A} \cdot \text{m}^{-2}$ in 3D bulk systems instead of the current with unit A in mesoscopic systems.

Next we show the detailed derivation from Eq.(11) to Eq.(12), particularly, how $\delta(\Omega_1)$ and $\delta(\Omega_1 - \omega)$ for the short time scale are obtained by recovering the time dependence of the second-quantization current operator and how $S_1^{a,(2)}$ and $S_2^{a,(2)}$ are related to the equal-time correlation calculated in our original manuscript. For brevity, we first suppress the superscripts (i) and (j) in Eq.(11) and obtain

$$\begin{aligned} S^{ab}(t, t') &= \frac{1}{2} \langle \Delta \hat{J}^a(t) \Delta \hat{J}^b(t') + \Delta \hat{J}^b(t') \Delta \hat{J}^a(t) \rangle_s \\ &= \frac{1}{2} \left(\langle \hat{J}^a(t) \hat{J}^b(t') \rangle_s - \langle \hat{J}^a(t) \rangle_s \langle \hat{J}^b(t') \rangle_s + \langle \hat{J}^b(t') \hat{J}^a(t) \rangle_s - \langle \hat{J}^b(t') \rangle_s \langle \hat{J}^a(t) \rangle_s \right), \end{aligned} \quad (18)$$

where $\hat{J}^a(t)$ is the second-quantization current operator derived above

$$\hat{J}^a(t) = \sum_{nm} \int_k J_{nm}^a(t) \hat{a}_n^\dagger \hat{a}_m, \quad (19)$$

where we have used the usual convention for the second-quantization operator. Straightforwardly, by substituting Eq.(19) into Eq.(18), we find:

$$S^{ab}(t, t') = \frac{1}{2} \sum_{mn} \sum_{m'n'} \int_k \int_{k'} J_{nm}^a(t) J_{n'm'}^b(t') \left[\langle a_n^\dagger a_m a_{n'}^\dagger a_{m'} \rangle_s - \langle a_n^\dagger a_m \rangle_s \langle a_{n'}^\dagger a_{m'} \rangle_s \right] \\ + \frac{1}{2} \sum_{mn} \sum_{m'n'} \int_k \int_{k'} J_{n'm'}^b(t') J_{nm}^a(t) \left[\langle a_{n'}^\dagger a_{m'} a_n^\dagger a_m \rangle_s - \langle a_{n'}^\dagger a_{m'} \rangle_s \langle a_n^\dagger a_m \rangle_s \right], \quad (20)$$

where $a_n^\dagger = a_n^\dagger(\mathbf{k})$, $a_m = a_m(\mathbf{k})$, $a_{n'}^\dagger = a_{n'}^\dagger(\mathbf{k}')$, $a_{m'} = a_{m'}(\mathbf{k}')$, $J_{nm}^a(t) = J_{nm}^a(\mathbf{k}; t)$, and $J_{n'm'}^b(t') = J_{n'm'}^b(\mathbf{k}'; t')$. At this stage, using¹²

$$\langle a_n^\dagger a_m a_{n'}^\dagger a_{m'} \rangle_s - \langle a_n^\dagger a_m \rangle_s \langle a_{n'}^\dagger a_{m'} \rangle_s = \delta(\mathbf{k} - \mathbf{k}') \delta(\mathbf{k} - \mathbf{k}') \delta_{mn} \delta_{nm'} f_n (1 - f_m), \quad (21)$$

we obtain

$$S^{ab}(t, t') = \frac{1}{2} \sum_{mn} \int_k J_{nm}^a(t) J_{mn}^b(t') [f_n (1 - f_m) + f_m (1 - f_n)] \\ = \frac{1}{2} \sum_{mn} \int_k J_{mn}^b(t') J_{nm}^a(t) [f_m (1 - f_n) + f_n (1 - f_m)] + \frac{1}{2} \sum_{mn} \int_k J_{mn}^b(t') J_{nm}^a(t) f_{nm}^2,$$

where the first term gives the thermal noise, which features the Fermi-surface property due to $f_n (1 - f_n) = -k_B T \partial f_n / \partial \epsilon_n$ and vanishes for gapped systems that we are interested in, while the second term gives the shot noise with $f_{nm} = f_n - f_m$ and features the Fermi-sea property. Note that $\delta(0)/V \equiv 1$ and $f_n (1 - f_m) + f_m (1 - f_n) = f_{nm}^2 + f_n (1 - f_n) + f_m (1 - f_m)$ have been used. Furthermore, by focusing on the autocorrelation ($a = b$), we find

$$S^{a,(i+j)}(t, t') = \frac{1}{2} \sum_{nm} \int_k J_{nm}^{a,(i)}(t) J_{mn}^{a,(j)}(t') f_{nm}^2, \quad (22)$$

where $S^a \equiv S^{aa}$ and the superscripts (i) , (j) , $(i + j)$ are used to indicate the order of the optical electric field. We comment that Eq.(22) is exactly the Eq.(4) in our original manuscript but with an explicit double time dependence.

The same as the DC photocurrent, the lowest-order DC shot noise is expected to be appeared at the second order of the optical electric field. Up to the second order of the optical electric field, the matrix element in Eq.(19) can be expanded as

$$J_{nm}^a(t) = J_{nm}^{a,(0)} + J_{nm}^{a,(1)}(t) + J_{nm}^{a,(2)}(t), \quad (23)$$

where $J_{nm}^{a,(0)} = v_{nm}^a$ is the zeroth order contribution which does not depend on time, $J_{nm}^{a,(1)}(t) = J_{nm}^{a,b\beta} e^{-i\omega_\beta t}$ and $J_{nm}^{a,(2)}(t) = J_{nm}^{a,b\beta c\gamma} e^{-i\omega_\Sigma t}$ are the first- and second-order contributions in the optical electric field, respectively, where $\omega_\Sigma \equiv \omega_\beta + \omega_\gamma$ and the explicit expressions for $J_{nm}^{a,b\beta}$ and

$J_{nm}^{a,b\beta c\gamma}$ without time dependence can be found in section B [particularly from Eq.(14), Eq.(16) and Eq.(17)] of our original Supplemental Material.

Next, from Eq.(22), the second-order shot noise $S_1^{a,(2)}(t, t')$ from the correlation of $J_{nm}^{a,(0)}$ and $J_{nm}^{a,(2)}$ is calculated as:

$$\begin{aligned} S_1^{a,(2)}(t, t') &= S_1^{a,(0+2)}(t, t') + S_1^{a,(2+0)}(t, t') \\ &= \frac{1}{2} \sum_{nm} \int_k f_{nm}^2 v_{mn}^a J_{mn}^{a,b\beta c\gamma} e^{-i\omega_\Sigma t'} + \frac{1}{2} \sum_{nm} \int_k f_{nm}^2 J_{nm}^{a,b\beta c\gamma} v_{nm}^a e^{-i\omega_\Sigma t} \\ &= \frac{1}{2} \sum_{nm} \int_k f_{nm}^2 v_{mn}^a J_{mn}^{a,b\beta c\gamma} \left(e^{-i\omega_\Sigma t} + e^{-i\omega_\Sigma t'} \right). \end{aligned} \quad (24)$$

By solving Eq.(13), we have

$$\begin{cases} t = t_0 + t_1/2, \\ t' = t_0 - t_1/2, \end{cases} \quad (25)$$

and hence we find

$$S_1^{a,(2)}(t_1, t_0) = \sum_{nm} \int_k f_{nm}^2 v_{mn}^a J_{mn}^{a,b\beta c\gamma} e^{-i\omega_\Sigma t_0} \cos(\omega_\Sigma t_1/2). \quad (26)$$

Taking average over t_0 in a period of $T \equiv 2\pi/\omega$, we find $\omega_\Sigma = 0$ or $\omega_\beta = -\omega_\gamma$ (This condition in fact gives the DC photocurrent) and

$$S_1^{a,(2)}(t_1) = \sum_{nm} \int_k f_{nm}^2 v_{mn}^a J_{mn}^{a,b\beta c\gamma}, \quad (27)$$

which is independent of t_1 . As a result, by performing a Fourier transform for the short time scale or time difference, we arrive at:

$$S_1^{a,(2)}(\Omega_1) = \delta(\Omega_1) \sum_{nm} \int_k f_{nm}^2 v_{mn}^a J_{mn}^{a,b\beta c\gamma} \equiv \delta(\Omega_1) S_1^{a,(2)} \quad (28)$$

where

$$S_1^{a,(2)} \equiv \sum_{nm} \int_k f_{nm}^2 v_{mn}^a J_{mn}^{a,b\beta c\gamma} \quad (29)$$

is the equal-time correlation between $J_{nm}^{a,(0)}$ and $J_{mn}^{a,(2)}$ evaluated in our original manuscript (differs a factor 1/2), which contains the shift and injection shot noise.

Following a similar way, $S_2^{a,(2)}(t, t')$ due to $J_{nm}^{a,(1)}(t)$ at different times is found to be

$$S_2^{a,(2)}(t, t') = \frac{1}{2} \sum_{nm} \int_k f_{nm}^2 J_{nm}^{a,b\beta} J_{mn}^{a,c\gamma} [e^{-i(\omega_\beta t + \omega_\gamma t')} + e^{-i(\omega_\beta t' + \omega_\gamma t)}]. \quad (30)$$

Using Eq.(25), we find

$$S_2^{a,(2)}(t_1, t_0) = \sum_{nm} f_{nm}^2 J_{nm}^{a,b\beta} J_{mn}^{a,c\gamma} e^{-i\omega_\Sigma t_0} \cos[(\omega_\beta - \omega_\gamma)t_1/2]. \quad (31)$$

Taking the averaging over t_0 , we also have $\omega_\Sigma = 0$ or $\omega_\beta = -\omega_\gamma$ and find

$$S_2^{a,(2)}(t_1) = \sum_{nm} \int_k f_{nm}^2 J_{nm}^{a,b\beta} J_{mn}^{a,c\gamma} \cos[\omega_\beta t_1], \quad (32)$$

which explicitly shows a time dependence on t_1 and hence belongs to the AC shot noise for short time scale. Taking the Fourier transform in t_1 time space, we arrive at

$$S_2^{a,(2)}(\Omega_1) = \frac{1}{2} \sum_{nm} \int_k f_{nm}^2 J_{nm}^{a,b\beta} J_{mn}^{a,c\gamma} [\delta(\Omega_1 + \omega_\beta) + \delta(\Omega_1 - \omega_\beta)] = \delta(\Omega_1 - \omega) S_2^{a,(2)} \quad (33)$$

where $\omega_\beta = \pm\omega$ has been used and

$$S_2^{a,(2)} \equiv \sum_{nm} \int_k f_{nm}^2 J_{nm}^{a,b\beta} J_{mn}^{a,c\gamma}, \quad (34)$$

is the equal-time autocorrelation from $J_{nm}^{a,(1)}$ evaluated in our original manuscript, which includes the shift, injection, jerk, and snap shot noise.

To summarize, by considering the time dependence of the optical electric field, we find that the second-order shot noise power can be expressed as $S^{a,(2)}(\Omega_1) = \delta(\Omega_1) S_1^{a,(2)} + \delta(\Omega_1 - \omega) S_2^{a,(2)}$. Here the first term due to the correlation of $\hat{J}^{a,(0)}$ and $\hat{J}^{a,(2)}$ indeed gives the shift and injection DC shot noise for both long and short time scales, while the second term due to the autocorrelation of $\hat{J}^{a,(1)}$ at different times gives an AC shot noise for the short time scale, in which the DC component for long time scale $(t + t')/2$ has been considered by taking the time average over a period T . In addition, due to the presence of $\delta(\Omega_1)$, we have an extra unit of time for the DC shot noise which has been missed in our original manuscript. In addition, we also missed a factor $1/V$ in \int_k in our original manuscript. We wish to thank the reviewer again for this very important question which enables us to identify the correct DC contribution of the second-order shot noise.

In the revised manuscript, we have rewritten (marked in blue) the section of [The quantum theory for DSNs], where we have calculated the double-time correlation by considering the time dependence of second-quantization photocurrent operator and have illustrated how the DC contribution is extracted. And the corresponding discussions in [INTRODUCTION] and [DISCUSSION] sections were revised, see their last paragraphs. In addition, we have corrected the unit of our *ab initio* calculations, where the thickness of 2D materials justified by Eq.(8) in the revised manuscript also be considered.

Concern 4: As a related issue, the present formulation of current operator in Eq.(2) could be used to compute equal time expectation value, but perhaps not applicable to compute the expectation value at different times which is necessary to evaluate $\langle J(t)J(0) \rangle$ and gives the main contribution to the shot noise. As a side note, Sipe and Shkrebtii (ref 17) is formulating photocurrent with the time evolution of the equal time expectation value, so it will be good to use this convention rather than writing like $\rho_{mn}^{(0)} = a_m^\dagger a_n$. Anyway, I suspect it is not straightforward to compute in this formalism.

Authors' reply: — We thank the reviewer for the comments and suggestions. As pointed out by the reviewer above, in our original manuscript, we have ignored the time dependence of the second-quantization current operator and only calculated the equal-time current correlation as the DC shot noise, this indeed is problematic. In the reply above, we have restored the time-dependence of the second-quantization current operator, which arises from the time-dependent optical field, and recalculated the current correlation at different times at the second order of optical field. Interestingly, after considering the time dependence in the second-quantization current operator, we find that the equal-time correlation in our original manuscript now are classified into two contributions, where the first one due to $\hat{J}^{a,(0)}$ and $\hat{J}^{a,(2)}$ gives the DC shot noise or $\Omega_1 = 0$ component of noise power for both long time scale and short time scale while the second one due to $\hat{J}^{a,(1)}$ at different times gives the AC shot noise for the short time scale. We hope that we have convinced the reviewer that our formulation can be used to calculate the DC shot noise spectrum or the $\Omega_1 = 0$ component of noise power for the short time scale or time difference $t - t'$, in which the DC component for long time scale $(t + t')/2$ has been considered by taking the time average over a period $T = 2\pi/\omega$.

Finally, we note that Sipe's formulation (Ref.2 of this reply) indeed has defined a second-quantization current density operator in Heisenberg picture, particularly by Eq.(17) in Ref.2,

$$\mathbf{J}(t) = \sum_{nm} \int \frac{d\mathbf{k}}{\Omega} \mathbf{v}_{nm} a_n^\dagger(\mathbf{k}; t) a_m(\mathbf{k}; t), \quad (35)$$

where Ω is same as the V in our formulation and the time dependence of a_n^\dagger and a_m is restored. In Heisenberg picture, $a_n^\dagger(t) = e^{iHt/\hbar} a_n^\dagger(0) e^{-iHt/\hbar}$ and $a_m(t) = e^{iHt/\hbar} a_m(0) e^{-iHt/\hbar}$, where we have suppressed \mathbf{k} in a_n^\dagger , H is the total Hamiltonian, and $a_n^\dagger(0)/a_m(0)$ is creation/annihilation operator in Schrödinger picture. In fact, Eq.(35) of this reply (or Eq.(17) in Ref.2) is equivalent to the second-quantization current operator used in our formulation. In particular, $a_n^\dagger(t) a_m(t)$ in Eq.(35) of this

reply is equivalent to the second-quantization density matrix element operator $\hat{\rho}_{nm} = \sum_i \hat{\rho}_{nm}^{(i)}$, where $\hat{\rho}_{nm}^{(0)} = a_n^\dagger(0)a_m(0)$ and $a_n^\dagger(0)/a_m(0)$ is the operator used in our formulation. To see that, we note that by requiring $\langle a_n^\dagger(t)a_m(t) \rangle_s = \frac{\Omega}{8\pi^3} c_{mn}$ (This equation is given above Eq.(30) of Ref.2), they also arrived at the Liouville equation for density matrix element c_{mn} ($e = \hbar = 1$):

$$\frac{\partial}{\partial t} c_{mn} + i\omega_{mn} c_{mn} = -c_{mn;b} E^b(t) + iE^b(t) \sum_p (r_{mp}^b c_{pn} - c_{mp} r_{pn}^b), \quad (36)$$

as given by Eq.(30) of Ref.2. This is because the operators that defined in different pictures after taking the average should give the same result. As a result, Eq.(35) (or Eq.(17) in Ref.2) exactly justify that our formulation is reasonable.

Concern 5: I also have a concern about the present formulation of shot noise based on the current operator in the bulk. J in Eq.(2) is the uniform $q=0$ component of the current. I am not sure this is the correct choice to evaluate the shot noise. Usually the shot noise is considered in a mesoscopic situation. For example, Souza et al [PRB 78, 155303 (2008)] treats shot noise by considering current between the contact and the system (not the current in the bulk) and derives Schottky formula $S=2eI$ using the Keldysh Green's function. As a check for validity to use the bulk current operator for defining shot noise, can the authors derive the Schottky formula $S=2eI$ by considering the metals, say 1 band system with a Fermi surface? I think this is crucial because if the present formulation cannot account for the conventional shot noise, the predicted shot noise with a relation to quantum geometry can be masked by additional shot noise from presence of the contact in a realistic situation, and perhaps not accessible in experiments.

Author's reply: We thank the reviewer for the insightful question. Generally speaking, the shot noise originates from the discrete nature of electric charge so that one can discuss it in mesoscopic systems as well as in bulk systems, where the former one has been well studied while the latter one is exactly formulated in this work based on photocurrent density operator instead of the current operator used in mesoscopic physics. As the very first step to investigate the shot noise caused by photocurrent (operator), we indeed only considered the $q = 0$ component of DC shot noise for short time scale or time difference. Here $q = 0$ may correspond two scenarios: (i) a uniform optical electric field is applied; and (ii) the bulk system is homogeneous, namely without contact and defect effects, which usually is the case of bulk photovoltaic effect. For the first case, our formulation based on nonlinear response theory may be extended to the $q \neq 0$ situation due to a nonuniform optical electric field since the current response for bulk systems driven by a

nonuniform electric field has been developed based on a similar theoretical method, see [PRB 99, 121111(R) (2019)] and [PRL 126, 156602 (2021)], which is an interesting and important topic but will be explored in future. In addition, the presence of the contact in a realistic situation usually can be safely ignored when it doesn't change the symmetry of bulk system, as will explained below. Finally, the defect effects in bulk systems is another important research topic, which may be explored in future.

Note that the shot noise theory for bulk systems developed in this work indeed shows many different characteristics from that discussed in mesoscopic systems. As a theoretical advancement, we need to compare both, which can be divided into the following two aspects, as suggested by the reviewer:

- (a). Is there a relation between the shot noise and the photocurrent in bulk systems like the relation $S = 2eI$ in mesoscopic systems?
- (b). Can the developed shot noise formulation for bulk systems that related to quantum geometry account for the conventional shot noise due to the inevitable contact effects in realistic experimental measurements?

Below we try to discuss these two aspects, mainly from the viewpoint of symmetry.

(a). In mesoscopic systems, the current and noise usually depend on voltage V in a nonlinear way. If we expand the current and noise in the linear bias regime, the ratio of the shot noise S and the current I is called Fano factor¹², given by

$$F = \frac{S}{2eI}. \quad (37)$$

Specifically, considering a two-terminal system with only one transmission channel, the current and shot noise linear in V are given by

$$I = \frac{e^2}{\pi h}TV, \quad S = \frac{2e^3|V|}{\pi h}T(1 - T), \quad (38)$$

respectively, where T is the transmission coefficient. By Eq.(38), it is found that $S = 2e(1 - T)I$. For the mesoscopic systems with a low transmission coefficient, we obtain the expression $S = 2eI$ or $F = 1$, which was first derived by Schottky and also studied in [PRB 78, 155303 (2008)] by Souza *et al.*. However, in the nonlinear regime such as the second-order nonlinear current and the second-order nonlinear shot noise, this ratio has not been studied. Therefore, we don't have a

reference definition like $S = 2eI$ in mesoscopic systems since the DC shot noise of photocurrent in bulk systems are second-order in the optical electric field.

In addition, sharply different from the shot noise discussed in mesoscopic systems, such as shot noise by considering current between the contact and the system discussed in [PRB 78, 155303 (2008)] by Souza *et al.*, we find that the symmetry in bulk systems plays an essential role for the DC shot noise for short time scale or time difference, the same as the DC photocurrent. For example, since both the DC shift and injection photocurrent susceptibility are \mathcal{P} -odd (\mathcal{P} : inversion symmetry) tensors, the \mathcal{P} -symmetry of bulk systems must be broken to capture these DC photocurrent responses. In addition, in a \mathcal{T} -invariant (\mathcal{T} : time reversal symmetry) bulk system, the shift photocurrent can only be excited by linearly polarized light (LPL) while the injection photocurrent can only be excited by circularly polarized light (CPL). Naturally, one can expect that the symmetry in bulk systems also constraints the DC shot noise. Particularly, our theoretical results show that DC shift (injection) shot noise can only be excited only by CPL (LPL) in \mathcal{T} -invariant bulk systems, which forms a dual dependence on the polarization of light with the DC photocurrent. As a result, by considering the symmetry constraints in bulk systems, one can not define the "Fano factor" or the Schottky relation between DC photocurrent I_{ph} and DC shot noise S_{ph} since S_{ph} and I_{ph} can hardly appear simultaneously for LPL and CPL.

(b). As discussed above, the symmetry for the DC shot noise in bulk systems plays a dominant role. We agree that the experimental measurements of DC shot noise may be affected by the inevitable contact effects, particularly when the contacts modify the symmetry of bulk systems. However, when the measurement setup doesn't change the symmetry of bulk systems, one can safely ignore the influence of contacts (including the possible shot noise) and argue that the theoretical formulation for bulk systems is predictable. (Usually, the contacts will lower the symmetry of bulk systems and hence allow responses that originally forbidden by symmetry of bulk systems, therefore the influence of contacts usually need to be minimized.) For example, the extrinsic second-order Hall effect in \mathcal{T} -invariant bulk systems predicted by Sodemann *et al.*⁵ based on the first-order semiclassical theory (also a theory for bulk systems), has been verified experimentally⁶⁻⁸, in which the influence of contacts can be safely ignored because it doesn't affect the symmetry of bulk system and the information of Berry curvature dipole (a geometric quantity) can be probed. Very recently, the intrinsic second-order anomalous Hall effect in \mathcal{PT} -symmetric bulk systems predicted by Gao *et al.*⁹ based on the second-order semiclassical theory (also a theory for bulk systems) has also been experimentally observed, where the influence of

contacts are discussed only by modifying the symmetry of bulk systems^{10,11} and the role of the quantum metric (a geometrical quantity) is revealed. Therefore, the shot noise or current fluctuation for bulk systems predicted in this work, that also related to the quantum geometry of bulk systems, could also be detected even when introducing the contacts.

In the revised manuscript, we have added a paragraph to compare our formulation with that in mesoscopic conductors, see the second paragraph of the [DISCUSSION] section.

In summary, we thank the reviewer for these constructive suggestions, which are very helpful in improving our manuscript. The modifications are marked in blue in the revised manuscript. We hope that the present manuscript is now suitable for publication in Nature Communications.

Reply to the reviewer #2:

Reviewer's comments: Authors describe how the DC shot noise (under the irradiation of light) in materials can be expressed in terms of geometric quantities (e.g., shift vector, quantum metric, and berry curvature). By examining the current operator (and separating it into shift and injection terms) they track various types of shot noise induced by light irradiation, delineating them into injection, shift, and snap noise contributions; they also provide a symmetry analysis and surprisingly find that the shot noise is not limited to inversion broken materials unlike the putative/claimed photocurrent counterpart; they finish their work with detailed simulations in GeS and MoS₂.

Authors' reply: We thank the reviewer for the careful reading and the positive comments on our manuscript. Below we will explain in detail why we claim that "the shot noise is not limited to inversion broken materials unlike the photocurrent counterpart".

Reviewer's comments: I find the paper addresses an interesting topic with potentially useful results. The results clearly should be published in some form but it is unclear if nature communications is the right venue. There are three key issues which probably need to be addressed:

Authors' reply: We thank the reviewer for the recommendation. Below we will address the issues raised by the reviewer.

Issue 1: [significance of authors results vs Ref 60] — Shot noise putatively/is claimed to be associated with photocurrent and specifically the shift photocurrent is not new. PRL 121, 267401 (Ref. 60) already discusses this. While it is clear that the authors have perhaps an updated and more general formulation (e.g., multi band vs two band), it is unclear to me whether and why this technical advance is so significant. For instance, authors say in footnote 60 that they have used a shift current operator (see page 2 for definition) as opposed to the local current operator of Ref 60. What insight does this provide? Are the authors saying that Ref. 60 results are wrong and that theirs is correct? As I understand it, all the shift current operator is (as shown on their page 2) is that they have just divided the current operator in to diagonal and off-diagonal contributions. Does this basically allow to delineate the contributions into various names (based on the diagonal/off-diagonal contributions) allowing them to name the contributions? If this is the basic difference, then why is the naming so important? Is it symmetry relations under T and PT symmetries? Authors should make this clear to exhibit their novelty better.

Authors' reply: We thank the reviewer for the critical comments about the novelty for our formulation by comparing with [PRL 121, 26740 (2018)]. Basically, this first issue raised by the reviewer can be organized into the following parts (Note that how the DSN is related to photocurrent will be explained in Issue 2):

- (i) *By comparing with [PRL 121, 26740 (2018)], what's the novelty of the current formulation, particularly for the shift shot noise by footnote 60 in our original manuscript? Is it a generalization of [PRL 121, 26740 (2018)] (from two band to multiband band)? If it is the case, why this technical advance is so significant?*
- (ii) *Using the shift current operator as opposed to the local current operator of [PRL 121, 26740 (2018)] also by footnote 60 in our original manuscript, what insight does this provide? Are the authors saying that the results of [PRL 121, 26740 (2018)] are wrong and that theirs is correct?*
- (iii) *As I understand it, all the shift current operator (as shown on their page 2) is that they have just divided the current operator in to diagonal and off-diagonal contributions. Does this basically allow to delineate the contributions into various names (based on the diagonal/off-diagonal contributions) allowing them to name the contributions If this is the basic difference, then why is the naming so important? Is it symmetry relations under \mathcal{T} and \mathcal{PT} symmetries?*

Below we address these questions one by one. For convenience, we will repeat the corresponding question listed above before giving the reply.

(i) By comparing with [PRL 121, 26740 (2018)], what's the novelty of the current formulation, particularly for the shift shot noise by footnote 60 in our original manuscript? Is it a generalization of [PRL 121, 26740 (2018)] (from two band to multiband band)? If it is the case, why this technical advance is so significant?

We are sorry for the inaccurate comments for [PRL 121, 26740 (2018)] in footnote 60 of our original manuscript. By carefully comparing our results with that of [PRL 121, 26740 (2018)], we find that they are fundamentally different, which can be seen from the following aspects:

- First of all, besides the optical electric field, an external (static) electric field E_{dc} in [PRL 121, 26740 (2018)] has been applied and the total current $J(E_{dc})$, given by Eq.(3) of [PRL

121, 26740 (2018)], is $J(E_{dc}) = J_{\text{shift}} + \sigma_E E_{dc}$, where the first term is the DC shift current in the absence of E_{dc} while the second term is an additional current induced by E_{dc} . However, the shot noise S calculated in [PRL 121, 26740 (2018)] has no contribution from J_{shift} as explicitly stated in [Annals of Physics 447, 169146 (2022)] written by the same authors: "... Note that S does not have the shot noise contribution proportional to current J_{shift} .", see the discussions below Eq.(12) of [Annals of Physics 447, 169146 (2022)]. Sharply different from that, our formulation allows us to probe the fluctuation of \hat{J}_{shift} , which has not been explored previously, as far as we know. Note that $\langle \hat{J}_{\text{shift}} \rangle_s = J_{\text{shift}}$, where $\langle \cdots \rangle_s$ stands for quantum statistical average and \hat{J}_{shift} is given by the off-diagonal contribution of $\sum_{nm} J_{O,nm}^{a,(2)}(t) a_n^\dagger a_m$. In addition, our formulation doesn't involve the external static electric field E_{dc} .

- Secondly, the shift shot noise discussed in our work is an intrinsic (free of relaxation time τ) effect while the shot noise discussed in [PRL 121, 26740 (2018)] is an extrinsic (depends on τ) effect and vanishes when $\tau = 0$, see Eq.(7) of [PRL 121, 26740 (2018)].
- Thirdly, possibly the most fundamental one, the formulation used by [PRL 121, 26740 (2018)] relies on the steady-state assumption, in which $S(t, t') = S(t - t')$, while our revised formulation are calculating the DC noise spectrum from $S(t, t') \neq S(t - t')$ since the second-quantization current operator in fact is time-dependent due to the dynamical optical electric field. We remark that the steady-state assumption adopted by [PRL 121, 26740 (2018)] has been clearly claimed in [Sci. Adv. 2016;2:e1501524] by T. Morimoto and N. Nagaosa: "... and (iii) a steady state is achieved.", see the last paragraph, left column, page 2 of [Sci. Adv. 2016;2:e1501524]. Note that the shot noise discussed in [PRL 121, 26740 (2018)] is based on the same theoretical framework.

Based on these facts, we can safely argue that our formulation is not a generalization of [PRL 121, 26740 (2018)] and naturally can not be regarded as a technical advance on top of [PRL 121, 26740 (2018)]. We are sorry again for the misleading comments in footnote 60 in our original manuscript. On the contrary, our work provides the very first formulation to probe the fluctuation of shift and injection current (we will explain this below) and stands for a novel theoretical advancement in nonlinear optics and also condensed matter physics.

(ii) *Using the shift current operator as opposed to the local current operator of [PRL 121, 26740*

(2018)] also by footnote 60 in our original manuscript, what insight does this provide? Are the authors saying that the results of [PRL 121, 26740 (2018)] are wrong and that theirs is correct?

In our formulation, the shift shot noise arises from the fluctuation of shift photocurrent operator \hat{J}_{shift} (whose quantum statistical average gives the shift photocurrent J_{shift}), while the shot noise discussed in [PRL 121, 26740 (2018)] based on local current operator is not directly related to J_{shift} . This is the insight that we originally want to provide by comparing the shift photocurrent operator with the local current operator in footnote 60 in our original manuscript. As explained above, the shot noise discussed in [PRL 121, 26740 (2018)] in fact is related to the current induced by an external static electric field, which doesn't include the contribution from shift photocurrent J_{shift} , while our formalism can describe the fluctuation of shift photocurrent operator. Although both our formulation and [PRL 121, 26740 (2018)] are discussing the shot noise in bulk systems, the problems addressed are fundamentally different. Therefore, we did not mean to say that [PRL 121, 26740 (2018)] is wrong by comparing the shift photocurrent operator used in our formulation with the local current operator used in [PRL 121, 26740 (2018)], but we are sorry for the misleading implication from footnote 60 in our original manuscript.

(iii) As I understand it, all the shift current operator (as shown on their page 2) is that they have just divided the current operator in to diagonal and off-diagonal contributions. Does this basically allow to delineate the contributions into various names (based on the diagonal/off-diagonal contributions) allowing them to name the contributions? If this is the basic difference, then why is the naming so important? Is it symmetry relations under \mathcal{T} and \mathcal{PT} symmetries?

In our formulation, the shift and injection photocurrent operators, respectively, are indeed defined by the off-diagonal ($\hat{J}_{\text{O}}^{a,(2)}$) and diagonal ($\hat{J}_{\text{D}}^{a,(2)}$) components of the second-order photocurrent operator ($\hat{J}^{a,(2)}$). But we didn't justify in detail why the $\hat{J}_{\text{O}}^{a,(2)}$ ($\hat{J}_{\text{D}}^{a,(2)}$) is related to the "shift" ("injection") contribution in our original manuscript, which in fact can be understood from the following three aspects:

- (a) The quantum statistical average of $\hat{J}_{\text{O}}^{a,(2)}$ and $\hat{J}_{\text{D}}^{a,(2)}$ gives the shift and injection current, respectively. Because of this, we call them shift and injection current operators, respectively.
- (b) As the first result of this division, the shift and injection currents (shot noises) have a clear symmetry implications as detailed below.

- (c) As the second result of this division, the calculated shot noise of shift and injection current operators have the same scaling of illumination time in short time limit as that of the shift and injection current, respectively.

For the first point, by taking the quantum statistical average for $\hat{J}_O^{a,(2)}$, in systems with time-reversal (\mathcal{T}) symmetry we find $\langle \hat{J}_O^{a,(2)} \rangle_s = 2\sigma^{abc} \text{Re}[E^b E^{c*}]$ with ($e = \hbar = 1$)

$$\sigma^{abc} = -\frac{i\pi}{4} \sum_{nm} \int \frac{d\mathbf{k}}{(2\pi)^d} f_{nm} (R_{mn}^{a,b} + R_{mn}^{a,c}) g_{nm}^{bc} \delta(\omega_{nm} - \omega), \quad (39)$$

where $g_{nm}^{bc} = r_{nm}^b r_{mn}^c + r_{nm}^c r_{mn}^b$ is the local quantum metric and $R_{mn}^{a,b} = -\partial_a \phi_{mn}^b + \mathcal{A}_m^a - \mathcal{A}_n^a$ is the shift vector, where $\partial_a = \partial/\partial k_a$, ϕ_{mn}^b is the phase factor of the interband Berry connection $r_{mn}^b = |r_{mn}^b| e^{i\phi_{mn}^b}$, and \mathcal{A}_n^a is the intraband Berry connection. Eq.(39) is exactly the shift photocurrent susceptibility tensor so that the quantum statistical average of $\hat{J}_O^{a,(2)}$ gives the shift photocurrent, namely $\langle \hat{J}_O^{a,(2)} \rangle = J_{\text{shift}}^a$. In this sense, $\hat{J}_O^{a,(2)}$ is defined as the shift photocurrent operator. We remark that the name of "shift" is arising from the shift vector, which determines the displacement of wave packet upon photoabsorption and leads to the shift photocurrent.

Similarly, by taking the quantum statistical average for $\hat{J}_D^{a,(2)}$, in \mathcal{T} -invariant systems we find $\langle \partial_t \hat{J}_D^{a,(2)} \rangle_s = 2\eta^{abc} \text{Im}[E^b E^{c*}]$ with

$$\eta^{abc} = -\frac{\pi}{2} \int \frac{d\mathbf{k}}{(2\pi)^d} \sum_{mn} f_{mn} \Delta_{mn}^a \Omega_{nm}^{bc} \delta(\omega_{nm} - \omega) \quad (40)$$

where $\Delta_{mn}^a = v_m^a - v_n^a$ is the group velocity difference of band m and n and $\Omega_{nm}^{bc} = i(r_{nm}^b r_{mn}^c - r_{nm}^c r_{mn}^b)$ is the local Berry curvature. Eq.(40) is exactly the injection photocurrent susceptibility tensor so that the quantum statistical average of $\hat{J}_D^{a,(2)}$ gives the injection photocurrent, namely $\langle \partial_t \hat{J}_D^{a,(2)} \rangle = \partial_t J_{\text{injection}}^a$. In this sense, $\hat{J}_D^{a,(2)}$ is defined as the injection photocurrent operator. Note that the injection current stems from the asymmetric motion of electrons and holes, as indicated by Δ_{mn}^a . Naturally, by calculating the quantum statistical average of photocurrent correlation operator $\hat{J}^{a,(0)} \hat{J}_O^a$ and $\hat{J}^{a,(0)} \hat{J}_D^a$, where $\hat{J}^{a,(0)} = \sum_{nm} \int_k v_{nm}^a a_m^\dagger a_n$, we obtain the DC shift and injection shot noise, respectively, as will be further explained in following reply, see reply (i) of Issue 2. Therefore, the division of $\hat{J}^{a,(2)}$ into off-diagonal and diagonal contributions for both current and shot noise indeed has a concrete physical insight.

In addition, as a result of this division, the obtained shift and injection photocurrents and noises indeed show a different dependence on symmetry, as suggested by the reviewer. For example, in \mathcal{T} -invariant systems, the shift shot noise due to $\hat{J}_O^{a,(2)}$ can only be excited by circularly polarized

light (CPL) while the injection shot noise due to $\hat{J}_D^{a,(2)}$ can only be excited by linearly polarized light (LPL). Interestingly, the shift (injection) current in \mathcal{T} -invariant systems can only be excited by LPL (CPL) that is complementary to its counterpart of DSN. Finally, also due to this division, we find that the shift current and the shift shot noise are independent on the illumination time while the injection current and the injection shot noise are linearly proportional to the illumination time in short time limit.

A further clarification: Note that in our original manuscript, we have only calculated the equal-time correlations and treated them as DC shot noise, this in fact is inaccurate, as pointed out by another reviewer. By considering the time dependence of current operator (arising from the time-dependent optical electric field), the final result given by Eq.(1) in our original manuscript should be changed as (see the section (II-B) of the revised Supplemental Material for details):

$$S^{a,(2)}(\Omega_1) = \delta(\Omega_1)S_1^{a,(2)} + \delta(\Omega_1 - \omega)S_2^{a,(2)}, \quad (41)$$

where the first term due to $\hat{J}^{a,(0)}(t)$ and $\hat{J}^{a,(2)}(t')$ is the DC shot noise for $t - t'$ due to $\delta(\Omega_1)$ and the second term due to $\hat{J}^{a,(1)}(t)$ at different times is the AC shot noise for $t - t'$ due to $\delta(\Omega_1 - \omega)$, where Ω_1 is the response frequency for $t - t'$. Interestingly, $S_1^{a,(2)}$ and $S_2^{a,(2)}$ in Eq.(41) are the equal-time correlations calculated in our original manuscript, but they can no longer be directly added up since they belong to different response frequency channels for $t - t'$. As explained above, by dividing $\hat{J}^{a,(2)}$ into off-diagonal and diagonal components, the corresponding photocurrent and shot noise are classified into shift and injection contributions, respectively. However, this classification strategy does not exist for $S_2^{a,(2)}$ since this division does not apply to the first-order photocurrent operator $\hat{J}^{a,(1)}(t)$. Besides this strategy, the photocurrent can still be classified as shift, injection, jerk and snap contributions by the scaling on illumination time t in short time limit, where $J_{\text{shift}} \propto t^0$, $J_{\text{injection}} \propto t^1$, $J_{\text{jerk}} \propto t^2$, and $J_{\text{snap}} \propto t^3$. It is by this classification strategy that the shot noise S_2 can be classified into shift, injection, jerk and snap contributions. In this work, we will only focus on the DC shot noise S_1 and hence the AC SN S_2 (was treated as DSN in our original manuscript) due to $\hat{J}^{a,(1)}$ at different times has been deleted.

In the revised manuscript (The revised parts in the revised manuscript and Supplemental material are marked in blue), (i) we remove the misleading comments in footnote 60 in our original manuscript and cited [PRL 121, 26740 (2018)] as Ref.38; (ii) we justify why the off-diagonal and diagonal contributions of $\hat{J}^{a,(2)}$ are named as the shift and injection photocurrent operators and also explain why the DC shot noises can be named as shift and injection ones, see the third para-

graph in the section of [The quantum theory for DSNs] in the revised manuscript; (iii) we show that the correlation function of $\hat{J}^{a,(1)}(t)$ at different times is of AC nature and hence we delete the detailed calculations, see subsections (II-A) and (II-B) of the revised Supplemental Material.

Issue 2: [is DSN really related to photocurrent?] — One striking thing in the authors work is that DSN is P even (see Table 1). This is really unusual since (A) DC rectified photocurrents arise from P broken materials and (B) authors claim that the DSN is related to the photocurrent generation processes. So I am confused: if the noise is there even for P even materials, why is it related to photocurrent at all? In the abstract, the authors propose that DSN is the quantum fluctuation of photocurrent. But if there is no photocurrent as forbidden by P symmetry why is this an OK characterisation? Surely, just because the current operator can be divided out into on-diagonal and off-diagonal contributions doesnt mean that they are shift or injection related? Perhaps authors could clarify. My suspicion is that the authors are actually probing shot noise under light irradiation whether or not photocurrent is generated. Can it be that this fluctuation is instead probing the optical absorption process (this is what injection and shift photocurrents ultimately stem from)? Note that optical absorption also scales with E^2 .

Authors' reply: We thank the reviewer for this insightful question and the contrustive suggestions. Basically, the second issues raised by the reviewer can be organized into the following parts:

- (i) *Why the shift/injection DSN in \mathcal{P} -invariant systems is related to shift/injection photocurrent that arises from \mathcal{P} -broken materials? And that the current operator can be divided into on-diagonal and off-diagonal contributions doesnt mean that they are shift or injection related?*
- (ii) *Why the shift/injection DSN is an OK characterisation for the fluctuation of shift/injection photocurrent in \mathcal{P} -invariant systems, where the photocurrent has been forbidden by \mathcal{P} -symmetry?*
- (iii) *Can it be that this fluctuation is instead probing the optical absorption process since the optical absorption also scales with E^2 ?*

Below we give a detailed explanation for these three parts. For convenience, we will repeat the corresponding question listed above before giving the reply.

(i) *Why the shift/injection DSN in \mathcal{P} -invariant systems is related to shift/injection photocurrent that*

arises from \mathcal{P} -broken materials? And that the current operator can be divided into on-diagonal and off-diagonal contributions doesn't mean that they are shift or injection related?

That the shift/injection current is nonzero only for \mathcal{P} -broken systems but the shift/injection DSN exists for \mathcal{P} -invariant system is counter-intuitive. We will address it below in question (ii) of this Issue. Here we focus on the second part of this question. We explained above that by dividing the second-order current operator into off-diagonal and diagonal components, we can obtain the shift and injection photocurrent, see reply (iii) of Issue 1. Next we explain in detail how the shift/injection DSN are related to the division of the second-order current operator. The second-order photocurrent operator can be written as

$$\hat{J}^{a,(2)}(t) = \underbrace{\sum_{nm} \int_k J_{\text{O},nm}^{a,(2)}(t) a_n^\dagger a_m}_{\hat{J}_{\text{O}}^{a,(2)}(t)} + \underbrace{\sum_{nm} \int_k J_{\text{D},nm}^{a,(2)}(t) a_n^\dagger a_m}_{\hat{J}_{\text{D}}^{a,(2)}(t)}, \quad (42)$$

where the quantum statistical average of the first term and the second term gives the DC shift and injection photocurrent, respectively. Then by evaluating the photocurrent correlation, we find that the shot noise of photocurrent at the second order of the optical electric field can be expressed as:

$$S^{a,(2)}(t, t') = \frac{1}{2} \sum_{nm} \int_k f_{nm}^2 \left[J_{nm}^{a,(0)}(t) J_{\text{O},mn}^{a,(2)}(t') + J_{nm}^{a,(0)}(t) J_{\text{D},mn}^{a,(2)}(t') \right], \quad (43)$$

where $J_{nm}^{a,(0)} = v_{mn}^a$ is independent of time. In Eq.(43), the first (second) term is defined as the shift (injection) shot noise since it contains $J_{\text{O},nm}^{a,(2)}$ ($J_{\text{D},nm}^{a,(2)}$). As a result, we find that the shift (injection) DSN encodes the same geometrical information as the shift (injection) photocurrent, such as the local quantum metric, the local Berry curvature. In addition, the DC shift (injection) photocurrent and shift (injection) DSN have the same scaling in illumination time, as discussed in Issue 1 (i-ii) above. In this sense, we argue that the shift (injection) DSN is related to the shift (injection) photocurrent, regardless of the \mathcal{P} -symmetry. Finally, we note that the presence of $J_{nm}^{a,(0)}$ in Eq.(43) leads to the different symmetry dependence between the shift (injection) DSN and the corresponding photocurrent. Note that the second-order correlation function due to $J_{nm}^{a,(1)}(t) J_{mn}^{a,(1)}(t')$ can not contribute to DSN because it belongs to AC component for short time scale, see reply (iii) of Issue 1).

(ii) Why the shift/injection DSN is an OK characterisation for the fluctuation of shift/injection photocurrent in \mathcal{P} -invariant systems, where the photocurrent has been forbidden by \mathcal{P} -symmetry?

Nonzero noise at zero current.

Phys. Rev. Lett. 75, 1610 (1995)

[REDACTED]

FIG. 5. Current noise and Fano factor. (a) S_i vs I curves, shifted vertically for visibility. The different colors represent the junctions' resistance values. The data follow the expected linear dependence in I . (b) Fano factor vs the normalized junction conductance (the panel contains more data than presented in panel (a)). Different colors indicate different Au atoms measured. The black line follows the theoretical prediction for a single spin-degenerate transport channel as is the case of gold. The inset shows the full scale of the theoretical prediction for a single transmission-channel junction.

Rev. Sci. Instrum. 93, 023702 (2022)

FIG. 1. The nonzero noise at zero current in mesoscopic conductors.

We totally agree with the reviewer that it is quite counterintuitive that the inversion symmetry forbids the DC photocurrent but allows for nonzero DC shot noise. In fact, when we speak of noise of photocurrent, we mean the correlation of photocurrent operator not the photocurrent itself, which can be seen from the general definition of noise spectrum:

$$S^{a,(i+j)}(t, t') \equiv \frac{1}{2} \langle \Delta \hat{J}^{a,(i)}(t) \Delta \hat{J}^{a,(j)}(t') + \Delta \hat{J}^{a,(j)}(t') \Delta \hat{J}^{a,(i)}(t) \rangle,$$

where $\Delta \hat{J}^{a,(i)} \equiv \hat{J}^{a,(i)} - \langle \hat{J}^{a,(i)} \rangle$. By this definition we can not say $S^{a,(i+j)} = 0$ when $\langle \hat{J}^{a,(i)} \rangle = 0$ and also $\langle \hat{J}^{a,(j)} \rangle = 0$. Interestingly, it has been well known that the current fluctuation in mesoscopic systems can be nonzero when the current is zero. For example, see FIG.1 of [Phys. Rev. Lett. 75, 1610 (1995)] and FIG.5 of [Rev. Sci. Instrum. 93, 023702 (2022)], which are reproduced as FIG.(1) in this reply. Therefore, the predicted shot noise without current can be an OK characterisation. We note that the zero photocurrent in \mathcal{P} -invariant systems is because that the left-going and right-going photocurrents cancel with each other due to \mathcal{P} symmetry. In addition, when we talk about the characterization of a physical quantity, e.g., the current, we mean to characterize the current operator. Hence its average, fluctuation, and higher order fluctuations are needed to fully characterize the current operator. Generally speaking, that the average is zero doesn't mean the fluctuation is also zero, as argued from the general definition of shot noise. Importantly, the fluctuation can also be informative. For instance, the shot noise of current in

mesoscopic systems can give information of the effect charge of the carriers as well as the quantum statistics (Fermi statistics or Bose statistics)¹²⁻¹⁶. And in this work, we want to employ the DSN to probe the geometrical information of quantum materials, regardless its \mathcal{P} -symmetry.

(iii) Can it be that this fluctuation is instead probing the optical absorption process since the optical absorption also scales with E^2 ?

We thank the reviewer for the constructive suggestions. As shown above, our formulation are tracking the fluctuation of the second-order current operator, whose quantum statistical average gives the shift and injection photocurrent, and in this sense we claim that our formulation are probing the fluctuation of photocurrent (operator). Note that the generation of DC photocurrent has included the optical absorption process, so is the DC shot noise. In fact, by illuminating the \mathcal{P} -symmetric gapped systems, the photoexcitation indeed has happened, but one can not detect the DC photocurrent since the \mathcal{P} -symmetry dictates that left-going photoexcited current will cancel out with the right-going photoexcited current. However, by applying a further static electric field, a "jerk" photocurrent¹⁷⁻¹⁹, which quadratically depends on the illumination time, will be generated even in \mathcal{P} -symmetric systems. Therefore, our DSN formulation offers a direct method to probe the correlation between the left-going and the right-going photoexcited currents, which also contains the shift and injection contributions.

In the revised manuscript, we have added the following discussion: "Intuitively, as a general feature of photocurrent, the \mathcal{P} -symmetry must be broken either by crystal structure or by external perturbation to guarantee that the left-going and right-going photocurrent cannot cancel with each other. However, for SN, this cancellation mechanism is lifted since the correlation of current is nonzero even when the current is zero, as exemplified by the notable Nyquist-Johnson noise in mesoscopic conductors.", see the second paragraph (marked in blue) of the section [The symmetry for DSNs].

Issue3: [probing DSN] How do the authors envision probing DSN? Given the DC character, do the authors envision a steady state experiment? If so, it would seem that relaxation should play a major role; a clearer explanation of this would have been nice (beyond the very short paragraph in the beginning of the discussion section). For instance, what happens when the carriers relax from the initial photo excitation energy to the band edge? Wouldnt the relaxation process play a role in the noise spectrum?

The measurement of DC shot noise

- (i) replace this with an insulating sample.
- (ii) illuminate the insulating sample by laser spot when measured.

Ultrafast photocurrent autocorrelation may be used.

[REDACTED]

Rev. Sci. Instrum. 93, 023702 (2022)

Nature Reviews Physics 5, 170–184 (2023)

FIG. 2. The possible experimental strategy to measure DSN.

Authors' reply: We thank the reviewer for this very important question. The experimental detection of DC shot noise may be performed with a setup similar to the one used in mesoscopic physics, for example, see the Block diagram shown in FIG.1 of Ref.[Rev. Sci. Instrum. 93, 023702 (2022)], in which one needs to replace the single-atom junction with an insulating sample, which will be illuminated by laser spot when measured, or see FIG.(2) of this reply, where the ultrafast photocurrent autocorrelation technique may be used²⁰. Note that in \mathcal{P} -symmetric systems, both the shift and injection photocurrent are forbidden (the left-going and the right-going photocurrent cancel with each other due to \mathcal{P} symmetry), therefore, to initiate the photocurrent correlation, when illuminating the light on the sample, an external static electric field should be applied, where a "jerk" photocurrent can be generated¹⁷⁻¹⁹. After that, by gradually decreasing the static electric field, a nonzero DSN signal at zero static electric field can be expected by our formulation.

In our formulation, the DSN is extracted from a double-time correlation function $S(t, t')$, so the relaxation processes (the photoexcited electrons lose their energy and relax to the conduction band edge of gapped systems) indeed play key roles, as suggested by the reviewer, particularly to distinguish different contributions (as justified by the paragraph in the beginning of the discussion section in our original manuscript). In the updated formulation, we in fact only have two DC shot

noises at the second order of the optical electric field, namely the shift and injection DSNs, which arise from the off-diagonal and diagonal contributions of $\hat{J}^{a,(2)}$, respectively, and therefore the relaxation processes for them should resemble the shift and injection photocurrent. Particularly, the shift photocurrent and DSN, which stand for intrinsic contributions, are less relevant to the impurity scattering. However, the injection photocurrent and DSN usually are related to the complicated scattering processes when relax to the edge of conduction band, and hence usually feature a longer relaxation time (about 10^{-12} to 10^{-14} s)²¹. Note that for the injection photocurrent, the relaxation process in most of studies is modeled by a constant relaxation time, such as in [Nature Communications 12, 4330 (2021)] and in [Nature Communications 10, 3783 (2019)], which is also the case for the injection DSN in this work. Besides classifying the contributions of photocurrent and DSN in terms of the time scale of relaxation process, we remark that the relaxation process may also be related to electron-phonon coupling or other coupling in realistic situation, generally $1/\tau = \frac{1}{\tau_{\text{phonon}}} + \frac{1}{\tau_{\text{impurities}}} + \dots$, where τ is the total relaxation time, but to figure out the detailed dependence is beyond the scope of the current work.

In this work, we formulated the theoretical framework for the first time to studying the shot noise of photocurrent. Obviously, it is only a starting point and may initiate the further study of quantum fluctuation by photocurrent. There are many important issues as pointed out by the reviewer, such as the relaxation process in the photoexcitation process, which will be the subject of future study. Finally, we remark that the short paragraph in the beginning of the discussion section in our original manuscript is delivered to justify the classification of the DSNs due to $\hat{J}^{a,(0)}$, $\hat{J}^{a,(2)}$ and $\hat{J}^{a,(1)}$, $\hat{J}^{a,(1)}$ on the equal footing (which has been used to classify the DC photocurrent), however, in the revised manuscript, we show that the second-order DSNs can only be contributed by $\hat{J}^{a,(0)}$, $\hat{J}^{a,(2)}$ and hence this classification has been deleted.

In the revised manuscript, we have added more discussions to the relaxation processes of DSNs, particularly in analogy with their photocurrent counterparts, see the first paragraph (marked in blue) of the section [DISCUSSION], and for the detection of DSNs, see the third paragraph (marked in blue) of the section [DISCUSSION], respectively.

In summary, we thank the reviewer for these constructive suggestions, which are very helpful in improving our manuscript. The modifications are marked in blue in the revised manuscript. We hope that the present manuscript is now suitable for publication in Nature Communications.

-
- ¹ C. Aversa and J.E. Sipe, Phys. Rev. B 52, 14636 (1995).
- ² J. E. Sipe and A. I. Shkrebtii, Phys. Rev. B 61, 5337 (2000).
- ³ Ya.M. Blanter and M. Büttiker, Phys. Rep. 1, 336 (2000).
- ⁴ R. A. Jishi, Feynman Diagram Techniques in Condensed Matter Physics (Cambridge University Press, 2013).
- ⁵ I. Sodemann and L. Fu, Phys. Rev. Lett. 115, 216806 (2015).
- ⁶ Q. Ma, S.-Y. Xu, H. Shen, D. MacNeill, V. Fatemi, T.-R. Chang, A. M. Mier Valdivia, S. Wu, Z. Du, C.-H. Hsu, S. Fang, Q. D. Gibson, K. Watanabe, T. Taniguchi, R. J. Cava, E. Kaxiras, H.-Z. Lu, H. Lin, L. Fu, N. Gedik, and P. Jarillo-Herrero, Nature **565**, 337 (2019).
- ⁷ S.-Y. Xu, Q. Ma, H. Shen, V. Fatemi, S. Wu, T.-R. Chang, G. Chang, A. M. M. Valdivia, C.-K. Chan, Q. D. Gibson, J. Zhou, Z. Liu, K. Watanabe, T. Taniguchi, H. Lin, R. J. Cava, L. Fu, N. Gedik, and P. Jarillo-Herrero, Nat. Phys. **14**, 900 (2018).
- ⁸ K. Kang, T. Li, E. Sohn, J. Shan, and K. F. Mak, Nat. Mat. **18**, 324 (2019).
- ⁹ Y. Gao, S. A. Yang, and Q. Niu, Phys. Rev. Lett. 112, 166601 (2014).
- ¹⁰ A. Gao *et al.*, Science, eadf1506 (2023).
- ¹¹ N. Wang *et al.* Nature, 1 (2023).
- ¹² Ya. M. Blanter and M. Büttiker, Phys. Rep. 1, 336 (2000).
- ¹³ M. Henny *et al.*, Science 284, 296 (1999).
- ¹⁴ M. P. Anantram and S. Datta, Phys. Rev. B 53, 16 390 (1996).
- ¹⁵ P. Samuelsson and M. Büttiker, Phys. Rev. Lett. 89, 046601 (2002).
- ¹⁶ S. Kolkowitz *et al.*, Science 347, 1129-1132 (2015).
- ¹⁷ B. M. Fregoso, R. A. Muniz, and J. E. Sipe, Phys. Rev. Lett. 121, 176604 (2018).
- ¹⁸ G. B. Ventura *et al.*, Phys. Rev. Lett. 126, 259701 (2021).
- ¹⁹ B. M. Fregoso, R. A. Muniz, and J. E. Sipe, Phys. Rev. Lett. 126, 259702 (2021).
- ²⁰ Q. Ma, R. K. Kumar, S.-Y. Xu, F. H. L. Koppens, and J. C. W. Song, Nat. Rev. Phys. 5, 170184 (2023).
- ²¹ A. M. Burger, R. Agarwal, A. Aprelev, E. Schrubba, A. Gutierrez-Perez, V. M. Fridkin, and J. E. Spanier, Sci. Adv. 5, eaau5588 (2019).

REVIEWER COMMENTS

Reviewer #1 (Remarks to the Author):

The authors greatly improved the manuscript by clarifying their formulation of shot noise. The convention of current expectation values is now clear, and the shot noise is correctly defined as a dc component of the current correlation function.

However, my concerns about the formulation are not fully resolved. Also I am not fully convinced whether the proposed shot noise of photocurrent is directly measurable. Below I list my questions and comments:

1. Below Eq. (2), the k integral (\int_k) is defined with $1/V$. The correct k summation is $\int dk/(2\pi)^d$ or $(1/V)\sum_k$ instead of $(1/V) \int dk/(2\pi)^d$. This unusual choice of k summation directly affects the dimension of the obtained shot noise. Can the authors clarify this point? For example, does the current expectation value $J(t)$ have the correct dimension of the current density?
2. The authors state that $S^{a,(2)}$ has the correct dimension for 3d. Do they also get the correct dimension for 1d systems?
3. Below Eq.(6) the connections of S to the geometric quantities are described in footnote 53. Since those connections are the main topic of the present works, it will be good if they are described in the main text (not in the footnote).
4. In the reply to concern 5 in my previous report, the authors discuss that the $S=2eI$ does not hold for their photocurrent DSN. I understand this and think it is an interesting aspect. But what I actually meant is: Can the conventional $S=2eI$ relationship for dc ohmic current (the current linear in the external dc electric field E) be reproduced by their bulk formulation for the shot noise? I asked this because it behaves as a criterion whether their new bulk formulation is justified for shot noise whose measurement inevitably involves the effect of electrodes.
5. I think the shot noise in Ref.[38] is not induced by the external electric field E_{dc} as the authors state in the reply, but by the ac electric field E that induces the photocurrent. It seems that S in [38] is related to the injection DSN η , although the approaches are different in terms of bulk vs local current operator.

On top of these comments, especially, I am not so sure if the proposed photocurrent DSN can be measurable in the experimental setups in the revised manuscript. In particular, the geometrical contributions have smaller powers of the relaxation time τ compared to the jerk and snap contributions and can be easily masked in real measurements. With these concerns about the novelty and experimental feasibility of the proposal, I do not recommend its publication in Nat. Commun. Rather I think the manuscript is suitable for more specialized journals after the above comments and questions are incorporated properly.

Reviewer #2 (Remarks to the Author):

It is clear that the authors have made significant improvements on their manuscript (e.g., new formulation of the shot noise to correct issues brought up by reviewer 1, etc.) believe that the manuscript should be published. I believe that nature communications is also an appropriate venue.

One thing that would have been nice to have in the manuscript is a fuller discussion of Ref. 38 and comparison of their work with Ref. 38. Now this discussion appears in the response to the reviewers. I believe explicitly stating the difference in the manuscript will help prevent further confusion in the community (this will be of service to future workers in the field).

Reply to the reviewer #1:

Reviewer's comments: The authors greatly improved the manuscript by clarifying their formulation of shot noise. The convention of current expectation values is now clear, and the shot noise is correctly defined as a dc component of the current correlation function. However, my concerns about the formulation are not fully resolved. Also I am not fully convinced whether the proposed shot noise of photocurrent is directly measurable. Below I list my questions and comments:

Authors' reply: We thank the reviewer for his/her comments about our work. Below we give a point-to-point reply to the concerns raised by the reviewer.

Reviewer's comments: 1. Below Eq. (2), the k integral (\int_k) is defined with $1/V$. The correct k summation is $\int dk/(2\pi)^d$ or $(1/V) \sum_k$ instead of $(1/V) \int dk/(2\pi)^d$. This unusual choice of k summation directly affects the dimension of the obtained shot noise. Can the authors clarify this point? For example, does the current expectation value $J(t)$ have the correct dimension of the current density?

Authors' reply: We thank the reviewer for this question. The factor $\frac{1}{V}$ in \int_k in our convention is related to the definition of the second-quantization photocurrent operator. Particularly, our formulation requires that

$$\langle a_m^\dagger(\mathbf{k})a_n(\mathbf{k}) \rangle_s = \delta(0)\delta_{nm}f_m, \quad (1)$$

where $\delta(0) = \delta(\mathbf{k} - \mathbf{k})$ has the dimension of $[L]^d$ with L and d the length and the spatial dimension, respectively. The $\delta(0)$ should be cancelled out by assuming $\delta(0) \times \frac{1}{V} = 1$, where the factor $\frac{1}{V}$ is absorbed into \int_k in our formulation. As a benchmark, we notice that Eq.(1) of this reply in fact is similar to the Eq.(16) of [Sipe's *et al.* PRB 61, 5337 (2000)], which is

$$\langle \Psi | a_n^\dagger(\mathbf{k})a_n(\mathbf{k}') | \Psi \rangle = \frac{\Omega}{8\pi^3} f_n \mathcal{D}(\mathbf{k} - \mathbf{k}'), \quad (2)$$

where $|\Psi\rangle$ and Ω stand for the ground state and the volume of the system, respectively, and $\mathcal{D}(\mathbf{k} - \mathbf{k}') = (1, 0)$ if $(\mathbf{k} = \mathbf{k}', \mathbf{k} \neq \mathbf{k}')$, which is dimensionless unlike $\delta(0)$ in our formulation. By comparing Eq.(1) of this reply with Eq.(2) of this reply or Eq.(16) of [Sipe's *et al.* PRB 61, 5337 (2000)], we find

$$\delta(0) \iff \Omega \mathcal{D}(\mathbf{k} - \mathbf{k}'). \quad (3)$$

Keep this in mind, we further note that the definition \int_k in our formulation in fact is similar to that of Eq.(17) of [Sipe's *et al.* PRB 61, 5337 (2000)], explicitly,

$$\int_k \equiv \frac{1}{V} \int \frac{d\mathbf{k}}{(2\pi)^d} \iff \int \frac{d\mathbf{k}}{\Omega}. \quad (4)$$

Note that the constant factor $\frac{1}{(2\pi)^d}$ with $d = 3$ in our formulation is placed to Eq.(2) of this reply or Eq.(16) of [Sipe's *et al.* PRB 61, 5337 (2000)].

It is the $\frac{1}{V}$ factor in \int_k as well as the Eq.(1) of this reply (that used in the Supplemental Material) that guarantee the correct dimension of the current density in Eq.(2) of our manuscript after taking quantum statistical average. For example, at zeroth order, we find $J_{mn}^{a,(0)} = v_{nm}^a$ so that

$$J^{a,(0)} \equiv \sum_{nm} \int_k J_{mn}^{a,(0)} \langle a_m^\dagger a_n \rangle_s = \sum_{nm} \frac{1}{V} \int \frac{d\mathbf{k}}{(2\pi)^d} v_{nm}^a \delta(0) \delta_{nm} f_n = \sum_n \int \frac{d\mathbf{k}}{(2\pi)^d} v_n^a f_n. \quad (5)$$

Note that we have assumed $\delta(0)/V \equiv 1$. By restoring e and \hbar , we find

$$[J^{a,(0)}] = \left[e \sum_n \int \frac{d\mathbf{k}}{(2\pi)^d} v_n^a f_n \right] = [Q] \times \frac{1}{[L]^d} \times \frac{[L]}{[T]} = \frac{[I]}{[L]^{d-1}}, \quad (6)$$

where $[I] = [Q]/[T]$ and Q , T , and I stand for the charge, time, and current, respectively. Eq.(6) of this reply gives the correct dimension for the current density, as expected. Finally, we note that the second-quantization photocurrent operator defined by Eq.(2) in our manuscript at second order of $\mathbf{E}(t)$ after quantum statistical average gives the same expressions for the well-known shift and injection photocurrents, see section (I-C) of the Supplemental Material, which also ensures that Eq.(2) in our manuscript gives the correct dimension for the current density. Similarly, by using Eq.(2) in our manuscript and the following identity¹

$$\langle a_n^\dagger a_m a_{n'}^\dagger a_{m'} \rangle_s - \langle a_n^\dagger a_m \rangle_s \langle a_{n'}^\dagger a_{m'} \rangle_s = \delta(\mathbf{k} - \mathbf{k}') \delta(\mathbf{k} - \mathbf{k}') \delta_{mn'} \delta_{nm'} f_n (1 - f_m), \quad (7)$$

we obtain the correct dimension of the DSN (DC shot noise).

Reviewer's comments: 2. The authors state that $S^{a,(2)}$ has the correct dimension for $3d$. Do they also get the correct dimension for $1d$ systems?

Authors' reply: We thank the reviewer for this question. In $1d$ systems, the current density has the dimension of $[I]$, the same as that of the current, therefore, the DSN in our formulation should give the dimension of $[I]^2$ for $1d$ systems. Next we show that our formulation indeed gives this

result. Particularly, the shift DSN susceptibility tensor by our formulation is given by [see Eq.(5) in our manuscript]

$$\sigma_C^{abc} = \frac{\pi}{4} \left(\frac{e^4}{\hbar^2} \right) \sum_{nm} \int_k f_{nm}^2 (W_{mn}^{abc} - W_{mn}^{acb}) \delta(\omega - \omega_{mn}), \quad (8)$$

where e and \hbar have been restored by dimension analysis [For zeroth (second) order current density, the universal constant is e and e^3/\hbar^2 so that we obtain the universal constant e^4/\hbar^2 for their quantum correlation] and $W_{mn}^{abc} = i(v_{mn;b}^a r_{nm;a}^c - v_{nm;b}^a r_{mn;a}^c)$, where $v_{mn;b}^a = \partial_b v_{mn}^a - (\mathcal{A}_m^b - \mathcal{A}_n^b) v_{mn}^a$ and $r_{mn;b}^a = \partial_b r_{mn}^a - (\mathcal{A}_m^b - \mathcal{A}_n^b) r_{mn}^a$ with $\partial_b \equiv \partial/\partial k_b$. With these notations, we have:

$$[S_{\text{sh}}^{a,(2)}] = [\sigma_C^{abc} E^b E^c] = \frac{[Q]^4}{[Q]^2 [\mathcal{V}]^2 [T]^2} \times \frac{1}{L^{2d}} \times \frac{[L]^2}{[T]} \times [L]^2 \times [T] \times \frac{[\mathcal{V}]}{[L]} \times \frac{[\mathcal{V}]}{[L]} = \frac{[I]^2}{[L]^{2d-2}}, \quad (9)$$

where \mathcal{V} stands for the voltage. For convenience, we have used different colors to build a quick connection between Eq.(8) and Eq.(9) of this reply for σ_C^{abc} . By Eq.(9), we find that the dimensions of the shift DSN are $[I]^2$, $[I]^2/[L]^2$, and $[I]^2/[L]^4$ for $1d$, $2d$, and $3d$ systems, respectively, as expected. The same conclusion is drawn for the injection DSN by a similar analysis.

Reviewer's comments: 3. Below Eq.(6) the connections of S to the geometric quantities are described in footnote 53. Since those connections are the main topic of the present works, it will be good if they are described in the main text (not in the footnote).

Authors' reply: We thank the reviewer for this constructive suggestion. In the revised manuscript, we have put the footnote 53 into the main text, see the text highlighted in blue below Eq.(5) and Eq.(6) in the revised manuscript.

Reviewer's comments: 4. In the reply to concern 5 in my previous report, the authors discuss that the $S = 2eI$ does not hold for their photocurrent DSN. I understand this and think it is an interesting aspect. But what I actually meant is: Can the conventional $S = 2eI$ relationship for dc ohmic current (the current linear in the external dc electric field E) be reproduced by their bulk formulation for the shot noise? I asked this because it behaves as a criterion whether their new bulk formulation is justified for shot noise whose measurement inevitably involves the effect of electrodes.

Authors' reply: We are sorry for misunderstanding the reviewer's question in our previous reply and we thank the reviewer for this important question. At the linear order of E , a similar relation

to $S = 2eI$ in principle can be derived using our bulk formulation under the DC limit. Particularly, by our formulation, the ratio (denoted as \mathcal{F}) between the first-order shot noise $S^{a,(1)}$ in DC electric field E^b and the first-order Ohmic current density $J^{a,(1)}$ is given by:

$$\mathcal{F} = \frac{S^{a,(1)}}{J^{a,(1)}} = \frac{\tau \frac{e^3}{\hbar} \frac{1}{2} \sum_{nm} \int_k f_{nm}^2 \Delta_{nm}^a g_{mn}^{ab} E^b}{\tau \frac{e^2}{\hbar} \sum_n \int \frac{d\mathbf{k}}{(2\pi)^d} f_n \partial_b v_n^a E^a} \equiv 2e\alpha, \quad (10)$$

where $\alpha = \sum_{nm} \int_k f_{nm}^2 \Delta_{nm}^a g_{mn}^{ab} E^b / [4 \sum_n \int d\mathbf{k} / (2\pi)^d f_n \partial_b v_n^a E^a]$ with g_{mn}^{ab} the quantum metric and Δ_{nm}^a the group velocity difference. As a comparison, the Fano factor derived by the scattering matrix theory for two-probe systems is given by

$$F \equiv \frac{S}{I} = 2e \frac{\sum_n T_n (1 - T_n)}{\sum_n T_n}, \quad (11)$$

where T_n is the transmission for the n th channel in mesoscopic conductors. For Eq.(11), we find $F = 2e$ when $T_n \ll 1$ and hence $S = 2eI$, as quoted by the reviewer. However, for our bulk formulation based on response theory, it is not obvious how to make a similar approximation to α such that $S^{a,(1)} = 2eJ^{a,(1)}$.

We realize that the difference between the two formulations is that the scattering matrix theory includes the effect of the electrodes while our bulk formulation does not. This in fact is the major concern raised by the reviewer, in our opinion. Interestingly, as will be discussed in detail below, it is with the insights from the scattering matrix theory about the effect of electrodes that we argue that the photocurrent DSNs predicted in our work can be measured in experiments that inevitably involve the effect of electrodes. In general, by the scattering matrix theory², the electron transport can be classified into the diffusive and the ballistic transport regimes, which can be justified by the contact resistance due to electrodes. For the former one, the effect of electrodes is minor (the contact resistance is much smaller than the bulk resistance) and therefore the predictions based on the bulk formulation (such as the response theory or the semiclassical theory) can be verified by the transport measurements; for the latter one, the effect of electrodes becomes important (the contact resistance can be comparable to the bulk resistance) and we argue that the predictions from the bulk formulation can still be extracted by using the symmetry. Next we elaborate on these arguments.

Within the scattering matrix theory, the Landauer formula for resistance is given by²:

$$R = \frac{h}{2e^2} \frac{1}{MT}, \quad (12)$$

where T represents the average transmission probability and M is the number of the available transport channels in the conductor. Eq.(12) of this reply shows that the scattering in the conductor is the source of the resistance since it reduces the transmission probability T .

However, when $T = 1$ (particularly in a ballistic conductor²) we still have a finite resistance

$$R_c = \frac{h}{2e^2} \frac{1}{M} \quad (13)$$

Where does this resistance come from? This puzzle was solved by Imry and the resistance is due to the electrodes^{2,3}, which are in contact with the conductor and hence R_c is called "contact resistance". By isolating the contribution of electrodes for the resistance, Eq.(12) can be recast into²

$$R = \frac{h}{2e^2 M} + \frac{h}{2e^2 M} \frac{1-T}{T} \equiv R_c + R_a, \quad (14)$$

where $R_a \equiv h(1-T)/(2e^2 MT)$ is the bulk or "actual" resistance caused by the scattering in the conductor. Particularly, for a diffusive conductor with a length L of $1\mu\text{m} \sim 10\mu\text{m}^2$, by taking $T = 0.01$ we have $R_c \simeq 1\%R_a$ and therefore the contact resistance or the effect of electrodes can be safely ignored. In fact, in a diffusive conductor, we have²:

$$R_a = \frac{L}{\sigma W}, \quad (15)$$

where σ is the conductivity and W is the width of the conductor. Importantly, R_a in the diffusive conductors obeys the Ohm's law $V = IR_a$ or $j = \sigma E$, where σ can be calculated from the bulk formulation, such as the response theory used in our work or the semiclassical theory⁴.

Taking the contact resistance as a ruler, we are ready to justify the influence of electrodes for our bulk formulation, which can be divided into two cases based on previous discussions:

- Diffusive transport regime;
- Ballistic transport regime.

Diffusive transport regime: In this regime, we learn from the scattering matrix theory that the effect of electrodes is less important and the Ohm's law $j = \sigma E$ holds, where σ can be derived from the bulk formulation such as the response theory or the semiclassical theory. As the nonlinear extensions of the Ohm's law, we note that the Berry curvature dipole (BCD) driven extrinsic response relation $j_a = \tau \sigma_1^{abc} E^b E^c$ predicted based on bulk semiclassical theory⁵ has been verified experimentally⁶. Very recently, the quantum metric dipole driven intrinsic response

[REDACTED]

Nature 565, 337–342 (2019). Nature 621, 487–492 (2023). Nat. Phys. 13, 842–847 (2017).

FIG. 1. The devices used to demonstrate that (a) the Berry curvature dipole (BCD) driven extrinsic nonlinear Hall effect, (b) the quantum metric driven intrinsic nonlinear Hall effect, and (c) the injection photocurrent. In all these experiments, the electrodes hold a width of μm .

relation $j = \sigma_2^{abc} E^b E^c$ based on the extended bulk semiclassical theory⁷ has also been verified experimentally^{8,9}. In addition, the response relation $j_a = \eta^{abc} |\mathbf{E} \times \mathbf{E}^*|$ for injection photocurrent has been used to detect the chirality of Weyl fermion in a topological semimetal TaAs¹⁰. In all these experiments, the geometric properties (such as BCD and quantum metric dipole) of Bloch electrons can be extracted, where the size of the conductor and the width of the electrodes is on the order of μm , as shown in FIG.1. Therefore, we argue that the bulk response equations $S^{a,(2)} = \sigma_C^{abc} |\mathbf{E} \times \mathbf{E}^*|$ as well as $S^{a,(2)} = \tau \eta_L^{abc} |\mathbf{E}(\omega)|^2$ proposed in our work can also be measured experimentally in a diffusive conductor (In addition to the metal, here the diffusive conductor also means an insulator under light illumination, where the electrons located at valence bands are excited to the conduction bands).

Ballistic transport regime: In this regime, the effect of electrodes is important, as manifested by the contact resistance in the scattering matrix theory. Even if the influence of the electrode can not be ignored, some important aspects that predicted by bulk formulation can still be extracted. For instance, based on the semiclassical theory, Sodemann and Fu⁵ predicted that the nonlinear Hall effect ($j_a = \tau \sigma_1^{abc} E^b E^c$) can exist in time-reversal invariant but inversion breaking systems, which has a quantum origin arising from the Berry curvature dipole (BCD). For 2D systems, it

was shown that the BCD (a pseudovector denoted as \mathbf{D}^5) vanishes for symmetries higher than a single mirror plane and the BCD is perpendicular to the mirror line. Therefore, the nonlinear Hall current is nonzero only if $\mathbf{D} \cdot \mathbf{E} \neq 0$. These predictions based on the bulk semiclassical theory have been verified experimentally⁶. Interestingly, based on a four-probe system with a C_4 symmetry for the configuration of electrodes but a lower symmetry for the scattering region Hamiltonian, Ref.[11] numerically arrives at the same conclusions by using the Keldysh formalism (equivalent to the scattering matrix theory) that includes the electrodes. Besides the BCD driven nonlinear Hall effect, Ref.[12] based on the same approach also obtained the conclusions for the Berry connection polarizability tensor (a band geometrical quantity) driven third-order nonlinear Hall effect, predicted by the bulk semiclassical theory and verified experimentally¹³.

In summary, although it is not easy to find a general approximation to reproduce the $S^{a,(1)} = 2eJ^{a,(1)}$ (similar to $S = 2eI$) with our bulk formulation, we argue that the geometric information from our bulk formulation can be read out in a diffusive conductor, where the effect of electrodes can be safely ignored with the insights from the scattering matrix theory. In the ballistic transport regime, where the effect of electrodes becomes important, we argue that some key information can also be extracted by using the symmetry, as exemplified by numerical calculations for nonlinear Hall effects with the Keldysh formalism. However, to have a quantitative understanding for our bulk noise formulation in a ballistic conductor is beyond current work, which is an interesting topic but will be explored in future. In the revised manuscript, we have properly added the above discussions, see the second paragraph (highlighted in blue) of [DISCUSSION] section in the revised manuscript.

Reviewer's comments: 5. I think the shot noise in Ref.[38] is not induced by the external electric field E_{dc} as the authors state in the reply, but by the ac electric field E that induces the photocurrent. It seems that S in [38] is related to the injection DSN η , although the approaches are different in terms of bulk vs local current operator.

Authors' reply: The statement that the shot noise is induced by the external electric field in Ref.[38] is, in fact, made by the same authors of Ref. [38], but in a later review article. Please see section [3.External bias and noise of shift current] of Ref.[14] of this reply. In Ref.[38], the current in the presence of an external dc electric field E_{dc} as well as an ac (light) electric field

$E(\omega)$ is given by Eq.(3) of Ref.[38], which is

$$J(E_{dc}) = J_{\text{shift}} + \sigma_E E_{dc}, \quad (16)$$

where the explicit expressions for J_{shift} and σ_E are given by the Eq.(4) and the Eq.(5) of Ref.[38], respectively, which are

$$J_{\text{shift}} = \frac{2\pi e^3}{\hbar^2 \omega^2} |E(\omega)|^2 \int [dk] \text{Im} \left[\left(\frac{\partial v}{\partial k} \right)_{12} v_{21} \right] \delta(\omega_{21} - \omega), \quad (17)$$

$$\sigma_E = \frac{4\pi e^4}{\hbar^3 \omega^2} |E(\omega)|^2 \tau^2 \int [dk] |v_{12}|^2 (v_{11} - v_{22}) R' \delta(\omega_{21} - \omega). \quad (18)$$

where σ_E is the coefficient of the current driven by E_{dc} . Here is the explanation for these equations given by Ref.[38]: "The physical meaning of the above expressions can be understood as follows. In our setup, the sample is subjected to constant light field, which produces the constant shift of the electrons associated with the interband transitions leading to the shift current J_{shift} . **With the dc electric field, the accelerated motion of the photoexcited electrons and holes generates additional current $\sigma_E E_{dc}$, which is proportional to the difference of the group velocities between the conduction and valence bands, i.e., $v_{11} - v_{22}$**"

On the other hand, the shot noise S is given by the Eq.(7) of Ref.[38], which is

$$S = \frac{e^4}{\hbar^2 \omega^2} E^2 \tau \int [dk] |v_{11} - v_{22}| |v_{12}|^2 \delta(\omega_{21} - \omega). \quad (19)$$

Why the authors of Ref.38 argue that S is due to $\sigma_E E_{dc}$ instead of J_{shift} ? We note that both S and σ_E contain the group velocity difference $v_{11} - v_{22}$ as well as the relaxation time τ (highlighted in blue color) while J_{shift} only due to the ac (light) electric field $E(\omega)$ does not include these elements, so it is natural to attribute S in Ref.[38] to $\sigma_E E_{dc}$. Although this conclusion does not appear explicitly in Ref.[38], it indeed appears explicitly in a later review written by one of the authors of Ref.[38]: "**... Note that S does not have the shot noise contribution proportional to current J_{shift} .**", see the last sentence in section **[3. External bias and noise of shift current]** of Ref.[14].

In addition, it seems that the Eq.(19) of this reply or the Eq.(7) of Ref.[38] is related to the injection DSN η in our formulation, because injection DSN in our formulation also includes τ as well as group velocity difference. However, by dimension analysis, we find S in Eq.(19) of this reply or the Eq.(7) of Ref.[38] gives a different dimension compared to our injection DSN. For example, in $1d$ systems, Eq.(19) gives the dimension of $[Q^2]/[T]$ [see the sentence below Eq.(S35)

in Ref.[38]'s Supplementary Material] while our injection DSN expression gives the dimension of $[Q]^2/[T]^2 = [I]^2$. More importantly, our injection DSN includes the local Berry curvature and the shift vector while the shot noise in Ref.[38] does not. Finally, we wish to mention that Ref.[38] is also a bulk formulation.

In our previous reply, we have carefully compared our formulation with that of Ref.[38] and we conclude that both formulations are fundamentally different, which can be summarized as: (i) Ref.[38] is based on steady-state assumption¹⁵ while our formulation does not assume that; (ii) Ref.[38] does not include the contribution from the shift photocurrent¹⁴ while our shift DSN is probing the fluctuation of shift photocurrent operator; (iii) Ref.[38] relies on an external DC electric field while our formulation does not involve that. Therefore, our work provides the very first theoretical formulation to probe the fluctuation of shift and injection current operator and stands for a novel theoretical advancement in nonlinear optics community as well as in condensed matter physics. In the revised manuscript, we attached these discussions to Ref.[38] in the revised manuscript.

Reviewer's comments: On top of these comments, especially, I am not so sure if the proposed photocurrent DSN can be measurable in the experimental setups in the revised manuscript. In particular, the geometrical contributions have smaller powers of the relaxation time τ compared to the jerk and snap contributions and can be easily masked in real measurements. With these concerns about the novelty and experimental feasibility of the proposal, I do not recommend its publication in Nat. Commun. Rather I think the manuscript is suitable for more specialized journals after the above comments and questions are incorporated properly.

Authors' reply: As explained above, our formulation is a novel theoretical advancement in nonlinear optics community as well as in condensed matter physics and provides the very first theoretical formulation to probe the fluctuation of shift and injection current operator, which relates to the band geometry of Bloch electrons like the other bulk response relations. Furthermore, with the insights from the scattering matrix about the effect of electrodes, we argue that our photocurrent DSNs based on response theory similar to other bulk current response relations can be measured in a diffusive conductor. Even in a ballistic conductor, some key information from bulk formulations can also be extracted. Finally, we wish to mention that there are only two DSNs at the second order of optical electric field in our revised manuscript, namely the shift and injection DSNs, the jerk and snap shot noises that feature the AC property discussed in our original manuscript have been

removed, see section [II-B: Eq.(48-52)] of the revised Supplemental Material. As for the shift and injection DSNs, they can be easily distinguished by the dependence on the polarization of incident light since for the shift and injection DSNs are mutually exclusive for a particular polarization.

In summary, we thank the reviewer for these constructive suggestions, which are very helpful in improving our manuscript again. The modifications are marked in blue in the revised manuscript. We hope that the present manuscript is now suitable for publication in Nature Communications.

Reply to the reviewer #2:

Reviewer's comments: It is clear that the authors have made significant improvements on their manuscript (e.g., new formulation of the shot noise to correct issues brought up by reviewer 1, etc.) believe that the manuscript should be published. I believe that nature communications is also an appropriate venue.

Authors' reply: We thank the reviewer for this recommendation.

Reviewer's comments: One thing that would have been nice to have in the manuscript is a fuller discussion of Ref.38 and comparison of their work with Ref.38. Now this discussion appears in the response to the reviewers. I believe explicitly stating the difference in the manuscript will help prevent further confusion in the community (this will be of service to future workers in the field).

Authors' reply: We thank the reviewer for this constructive suggestion. In the revised manuscript, we have properly attached the discussions in response letter to Ref.38 in the revised manuscript.

-
- ¹ Ya. M. Blanter and M. Büttiker, *Phys. Rep.* 1, 336 (2000)
 - ² S. Datta, *Electronic Transport in Mesoscopic Systems*, (Cambridge, University Press 1995).
 - ³ Y. Imry, "Physics of mesoscopic systems" in *Directions in Condensed Matter Physics*, eds. G. Grinstein and G. Mazenko (World Scientific Press, Singapore, 1986).
 - ⁴ C.-P. Zhang, X.-J. Gao, Y.-M. Xie, H. C. Po, and K. T. Law, *Phys. Rev. B* 107, 115142 (2023).
 - ⁵ I. Sodemann and L. Fu, *Phys. Rev. Lett.* 115, 216806 (2015).
 - ⁶ Q. Ma *et al.*, *Nature* 565, 337–342 (2019).
 - ⁷ Y. Gao, S.Y.A. Yang, and Q. Niu, *Phys. Rev. Lett.* 112, 166601 (2014).
 - ⁸ A. Gao *et al.*, *Science* 381(6654): eadf1506 (2023).
 - ⁹ N. Wang *et al.* *Nature* 621, 487–492 (2023).
 - ¹⁰ Q. Ma *et al.*, *Nat. Phys.* 13, 842–847 (2017).
 - ¹¹ M.M. Wei, B. Wang, Y.J. Yu, F.M. Xu, and J. Wang, *Phys. Rev. B* 105, 115411 (2022).
 - ¹² M.M. Wei, L.J. Xiang, L.Y. Wang, F.M. Xu, and J. Wang, *Phys. Rev. B* 106, 035307 (2022).
 - ¹³ S. Lai, H. Liu, Z. Zhang, J. Zhao, X. Feng, N. Wang, C. Tang, Y. Liu, K. S. Novoselov, S. A. Yang, and W.-B. Gao, *Nat. Nanotechnol.* 16, 869 (2021).
 - ¹⁴ N. Nagaosa, *Ann. Phys.* 447, 169146 (2022).
 - ¹⁵ T. Morimoto and N. Nagaosa, *Sci. Adv.* 2, 1501524 (2016).

REVIEWER COMMENTS

Reviewer #1 (Remarks to the Author):

I appreciate the authors' clarifications including those on the unit of their current noise. Yet I strongly disagree with their assessment of Ref. 38.

1. The noise formula Eq. (7) in Ref. 38 describes the current noise induced by ac electric field *in the absence of* dc electric field E_{dc} . Namely E in Eq. (7) denotes the ac electric field $E(\omega)$, not the dc electric field E_{dc} . When the I-V characteristic is discussed in the former half of the paper, they indeed include the additional dc electric field E_{dc} , but in the latter half, the current noise is considered without the additional dc electric field. Actually this is clear from the fact that the noise formula should include the ac electric field $E(\omega)$ in some form (otherwise current (and its noise) should be zero for insulators.) This is also clear by looking at the discussions below Eq. S27 in the SM, where they compute current noise with the Floquet two band model in Sec. SI which only incorporates the ac electric field (not dc electric field). Thus the statement in footnote [38] in the present manuscript "(iii) their results rely on an external DC electric field while our formulation does not involves that" is incorrect and should be removed.

2. Also the statement "(ii) their results do not include the contribution from the shift photocurrent [39] while our shift DSN is probing the fluctuation of shift photocurrent operator." is a misunderstanding. Basically Eq. (7) in Ref. 38 is also computing the photocurrent fluctuation under the light irradiation. I think the statement in the later review Ref. 14 is made in the context that the current noise from the shift current does not involve a contribution in the form $S=2eJ_{\text{shift}}$ in contrast to the conventional shot noise $S=2eJ_{\text{ohmic}}$ for Ohmic current. As in the discussions below Eq. (7) in Ref. 38, the current noise of the shift current does not include the contribution that is directly proportional to the photocurrent J_{shift} , which makes a sharp contrast with the conventional shot noise $S=2eJ$ for Ohmic current.

3. Thus the setup for photocurrent noise is basically the same for Ref. 38 and the present work. The difference is in the approach that those two works adopt; Ref. 38 computes local current in the steady state while the present manuscript is based on the bulk uniform current. The difference of the unit $[Q^2]/[T^2]$ vs $[Q^2]/[T]$ comes from the fact that the authors extract the delta function $\delta(\omega=0)$ in front in their expression. By broadening the delta function with the width $1/\tau$ (τ : the relaxation time), I think these two expressions at least have the same unit.

4. In addition, I think the local current approach in Ref. 38 can reproduce the shot noise $S=2eJ$ once it is applied to Ohmic current. Let us suppose that one substitutes $G<$ and $G>$ in Eq. S27 in SM with the Green's functions under (only) the dc electric field. For example, this can be done by using the gauge invariant formulation in Sec. SIIA. Then I think it is straightforward to obtain the usual shot noise formula. This makes me feel afraid that the authors' formulation of the current fluctuation from the bulk uniform current operator may miss some important contributions. But as they argue, looking at the diffusive regime will be okay to justify their treatment.

In this regard, I believe the present work is not "fundamentally different" from Ref. 38 as they write in [38], but rather they pursued the photocurrent noise along the same direction with Ref.38. Combining

this fact with expected difficulty to measure the geometrical component of the noise (I don't think it's fair to compare the current noise measurement with direct current measurement like nonlinear hall effect.), I still think the manuscript is more suitable for publication in a more specialized journal after suitably revising the statement regarding the relationship with Ref. 38.

Reviewer #3 (Remarks to the Author):

Shot noise, a direct consequence of charge quantization, provides a unique lens to peer into quantum materials beyond what traditional conductance measurements offer. Specifically, shot noise experiments enable the determination of the charge and statistics of quasiparticles involved in transport, shedding light on the potential profile and internal energy scales within mesoscopic systems. This article discusses using light to detect the response of quantum materials, presenting a novel approach that involves the quantum fluctuation of photocurrent, specifically in the form of DC shot noise (SN), for the exploration of geometric properties in materials. The authors systematically develop a quantum theory and DFT calculations for DC shot noise, revealing and categorizing the resonant shift and injection DC shot noises. Additionally, they demonstrate that the expressions defining the shot noise susceptibility tensors are intricately linked to gauge-invariant geometric quantities. This is a well-written and timely article that makes a significant contribution to the field of quantum materials research. The authors' proposal to use DC shot noise as a probe for geometric properties is promising, and their theoretical framework is well-developed and convincing. This approach could be particularly valuable for materials that are difficult to probe using traditional methods. Overall, this is an excellent article that I recommend for publication. Undoubtedly, this work will captivate a diverse audience within the realm of quantum materials physics. Below, I list my comments and questions:

1. Firstly, reviewer #1 raised a pivotal query concerning the measurement of shot noise, prompting a need for a more detailed discussion by the authors on the setup of experimental verifications.
2. In the context of an inversion-symmetric system with time reversal symmetry, where Berry curvature and shift vector vanish, the dominance of the quantum metric in shot noise becomes significant. Further exploration of this phenomenon, particularly in centrosymmetric systems where the 2nd order DC current response is absent, would be valuable for discussion.
3. Regarding Figure 1 (d), the k-resolved distribution exhibits positivity at $\hbar\omega=2.5\text{eV}$ throughout the Brillouin zone, but below 2eV , a negative value emerges in the mmm phase. The authors are encouraged to elucidate the underlying physics driving this behavior.
4. Notably, it appears that the authors did not incorporate spin-orbit coupling (SOC) in the DFT calculations. Given the substantial SOC in MoS₂, an inclusion of a discussion on the influence of SOC on shot noise would enhance the completeness of the study.

Reply to the reviewer #1:

Reviewer's comments: I appreciate the authors clarifications including those on the unit of their current noise. Yet I strongly disagree with their assessment of Ref. 38.

Authors' reply: The assessment of Ref. 38 in the previous draft was partially based on the opinion of one of the authors of Ref. 38. We have modified the assessment of Ref. 38 in the revised version and here is our detailed reply to the comments of the reviewer.

Reviewer's comments: 1. The noise formula Eq. (7) in Ref. 38 describes the current noise induced by ac electric field *in the absence of* dc electric field E_{dc} . Namely E in Eq. (7) denotes the ac electric field $E(\omega)$, not the dc electric field E_{dc} . When the I-V characteristic is discussed in the former half of the paper, they indeed include the additional dc electric field E_{dc} , but in the latter half, the current noise is considered without the additional dc electric field. Actually this is clear from the fact that the noise formula should include the ac electric field $E(\omega)$ in some form (otherwise current (and its noise) should be zero for insulators.) This is also clear by looking at the discussions below Eq. S27 in the SM, where they compute current noise with the Floquet two band model in Sec. SI which only incorporates the ac electric field (not dc electric field). Thus the statement in footnote [38] in the present manuscript (iii) their results rely on an external DC electric field while our formulation does not involves that is incorrect and should be removed.

Authors' reply: We agree with the reviewer that the electric field explicitly appeared in Eq.(7) of Ref. 38 is the AC electric field. Specifically, Eq.(7) of Ref. 38 is given by

$$S = \frac{e^4}{\hbar^2 \omega^2} E^2 \tau \int [dk] |v_{11} - v_{22}| |v_{12}|^2 \delta(\omega_{21} - \omega), \quad (1)$$

where E is the AC electric field. Since the shift current has the following features: (1). it is intrinsic free of the relaxation time τ ; (2). it contains shift vector. Therefore, it is expected that the noise due to the shift current fluctuation also contains these features, as obtained in our formulation. However, S in Eq.(1) is proportional to τ and the group velocity difference $v_{11} - v_{22}$, which does not appear in the shift current given by Eq.(4) of Ref. 38 (which features the intrinsic nature and contains the shift vector). In contrast, both τ and $v_{11} - v_{22}$ appear in the current $\sigma_E E_{dc}$ that induced by both the AC electric field (included in σ_E) and the external dc electric field, see Eq.(5) of Ref. 38. In this sense, S in Ref. 38 relies on an external dc electric field, although E_{dc} does not explicitly appear in Eq.(7). Interestingly, as pointed out by the reviewer in the previous

report, Eq.(1) seems to resemble the injection DSN η in our formulation for injection current, but which has not been taken into account by Ref. 38.

In the revised manuscript, we have removed assessment (iii) in Ref. 38 since E_{dc} does not explicitly appear in Eq.(7) of Ref. 38. However, even without this assessment (iii), our formulation is still different from Ref. 38, as will be explained below.

Reviewer’s comments: 2. Also the statement (ii) their results do not include the contribution from the shift photocurrent [39] while our shift DSN is probing the fluctuation of shift photocurrent operator. is a misunderstanding. Basically Eq. (7) in Ref. 38 is also computing the photocurrent fluctuation under the light irradiation. I think the statement in the later review Ref. 14 is made in the context that the current noise from the shift current does not involve a contribution in the form $S = 2eJ_{shift}$ in contrast to the conventional shot noise $S = 2eJ_{ohmic}$ for Ohmic current. As in the discussions below Eq. (7) in Ref. 38, the current noise of the shift current does not include the contribution that is directly proportional to the photocurrent J_{shift} , which makes a sharp contrast with the conventional shot noise $S = 2eJ$ for Ohmic current.

Authors’ reply: In essence, the usual $S = 2eI$ relation reveals that there must be some relation between the current and its fluctuation. And this relation holds only in the **linear regime**, as discussed in Büttiker’s review Ref.[1] as well as in previous reply. Hence, the relation $S = 2eJ_{shift}$ does not hold in Ref.38 since J_{shift} is a nonlinear current. However, we indeed can expect that there should be some other connections between S and J_{shift} when S is contributed by J_{shift} . Interestingly, Eq.(7) of Ref. 38 does not seem to relate to J_{shift} , although both are induced by optical field. In particular, Eq.(7) of Ref. 38 does not contain the key quantity—**shift vector**, which is believed to be the physical origin of shift current². However, our formulation explicitly contains this quantity, see discussions below Eq.(5) in our manuscript. The reason behind this discrepancy may be that the local current operator does not contain the information of shift current, which is root in the off-diagonal of the velocity matrix element³. In addition, Eq.(7) of Ref. 38 depends on the relaxation time while the shift current does not depend it. As a comparison, our shift DSN contains the shift vector and does not depend on the relaxation time.

In the revised manuscript, we have modified the assessment (ii) as: **(ii) their shot noise formula does not contain the key geometric quantity—shift vector, which is believed to be the physical origin of shift current, while our formulation explicitly contains that quantity.**

Reviewer’s comments: Thus the setup for photocurrent noise is basically the same for Ref. 38 and

the present work. The difference is in the approach that those two works adopt; Ref. 38 computes local current in the steady state while the present manuscript is based on the bulk uniform current. The difference of the unit $[Q^2]/[T^2]$ vs $[Q^2]/[T]$ comes from the fact that the authors extract the delta function $\delta(\omega = 0)$ in front in their expression. By broadening the delta function with the width $1/\tau$ (τ : the relaxation time), I think these two expressions at least have the same unit.

Authors' reply: Based on the assessments (i) and (ii), we respectfully do not agree that our work is the same as Ref. 38. Furthermore, our formulation includes both the shift (intrinsic) and injection (extrinsic) DSNs, while Ref. 38 only contains one extrinsic DSN. Interestingly, the formulation of Ref.38 can only give the shift current. It does not contain the injection current and thereby it may not be used to derive the injection DSN. As a comparison, the shift and injection DSNs in our formulation are derived on the same footing.

In addition, our shot noise formula does not has the same unit as the Ref. 38, even for the one-dimensional case that the reviewer has examined. To see that, we note that the dimension of the shift DSN $S_{\text{sh}}^{a,(2)}$ in our formulation is given by [see Eq.(9) in our previous reply]

$$[S_{\text{sh}}^{(a,2)}] = \frac{[I]^2}{[L]^{2(d-1)}} = \frac{[Q]^2}{[T]^2[L]^{2(d-1)}}, \quad (2)$$

where I is the current, L the length, d the spatial dimension, Q the charge, and T the time. Note that $S^{a,(2)}$ has the unit of J^2 with J being the current density in any dimension. However, for Eq.(7) of Ref. 38, by dimension analysis we find:

$$S = \frac{e^4}{\hbar^2} \times \frac{1}{\omega^2} \times E^2 \times \tau \times \int [dk] \times |v_{11} - v_{22}| |v_{12}|^2 \times \delta(\omega_{21} - \omega),$$

$$\Rightarrow [S] = \frac{[Q]^4}{[Q]^2[\mathcal{V}^2][T]^2} \times [T]^2 \times \frac{[\mathcal{V}]^2}{[L]^2} \times [T] \times \frac{1}{[L]^d} \times \frac{[L]^3}{[T]^3} \times [T] = \frac{[Q]^2}{L^{(d-1)}[T]}, \quad (3)$$

where \mathcal{V} is the voltage. It is easy to find that $[S_{\text{sh}}^{a,(2)}]/[S] = 1/[L]^{d-1}/[T]$. In particular, for different spatial dimensions, we find:

- $d = 1$: $[S_{\text{sh}}^{(a,2)}]$ vs $[S] \iff [Q]^2/[T]^2$ vs $[Q]^2/[T]$ (This is examined by the reviewer);
- $d = 2$: $[S_{\text{sh}}^{(a,2)}]$ vs $[S] \iff [Q]^2/[T]^2/[L]^2$ vs $[Q]^2/[L]/[T]$;
- $d = 3$: $[S_{\text{sh}}^{(a,2)}]$ vs $[S] \iff [Q]^2/[T]^2/[L]^4$ vs $[Q]^2/[L]^2/[T]$.

Hence, the reviewer's arguments for $d = 1$ can not explain the difference of the unit for $d = 2$ and $d = 3$ cases, where the length L plays a role. Note that $\delta(\Omega_1)$ in Eq.(1) of our manuscript

arises from the Fourier transformation for the short time scale, which extracts the DC or the zero-frequency component of the shot noise, see the discussions below Eq.(4) in our manuscript.

Finally, we wish to emphasize that the unit for DSN in our formulation always be related to the square of current density for $d = 1, 2,$ and 3 . Therefore, our formulation displays a clear physical meaning— it quantifies the fluctuation of current density in any spatial dimension. However, Eq.(7) of Ref. 38 can not give a meaningful unit, namely how to relate the current density or the current. Note that the shot noise discussed in mesoscopic conductors has the unit of $[I]^2 = [Q]^2/[T]^2$ since they are evaluating the current fluctuation instead of the current density fluctuation. And this can also been seen from the usual $S = 2eI$ relation, where a factor $\delta(\omega)$ before S usually is ignored, see the equation below Eq.(49) of Ref.[1]. For convenience, we reproduce it as follows:

$$2\pi\delta(\omega + \omega')S_{\alpha\beta}(\omega) = \langle \Delta\hat{I}_\alpha(\omega)\Delta\hat{I}_\beta(\omega') + \Delta\hat{I}_\beta(\omega')\Delta\hat{I}_\alpha(\omega) \rangle, \quad (4)$$

where \hat{I}_α is the current operator with the dimension $[Q]/[T]$. In summary, our formulation is not the same as Ref. 38 but presents a novel theoretical advancement in nonlinear optics and condensed matter physics.

Reviewer's comments: 4. In addition, I think the local current approach in Ref. 38 can reproduce the shot noise $S = 2eJ$ once it is applied to Ohmic current. Let us suppose that one substitutes $G^<$ and $G^>$ in Eq. S27 in SM with the Greens functions under (only) the dc electric field. For example, this can be done by using the gauge invariant formulation in Sec. SIIA. Then I think it is straightforward to obtain the usual shot noise formula. This makes me feel afraid that the authors formulation of the current fluctuation from the bulk uniform current operator may miss some important contributions. But as they argue, looking at the diffusive regime will be okay to justify their treatment.

Authors' reply: The local current operator used in Ref. 38 is given by

$$v = \frac{1}{2L} \sum_{k,k'} (v_k + v'_k) c_k^\dagger c_{k'}, \quad (5)$$

which depends on the spatial dimension L and thereby in fact is current density operator. As a consequence, the correlation between the local current operators should also depend on the spatial dimension. So we think that the local current operator approach can not reproduce $S = 2eI$ since it does not depends on spatial dimension like our formulation. Furthermore, we think that $S = 2eI$ for Ohmic current can not affect the DSN proposed in our work. Particularly, our

formulation is focusing on insulating systems, where the linear Ohmic current J_{Ohmic} in metallic systems solely driven by a DC electric field can not appear and therefore $S = 2eJ_{\text{Ohmic}} = 0$ (shot noise at linear order) in our setup. As a consequence, the second-order DSN proposed in our work could be the leading contribution. Note that the possible $S = 2eI$ noises in metallic electrodes inevitably appeared in the conventional noise experimental measurements can also be bypassed by the noninvasive experimental technique, as discussed below.

Reviewer's comments: In this regard, I believe the present work is not fundamentally different from Ref. 38 as they write in [38], but rather they pursued the photocurrent noise along the same direction with Ref.38. Combining this fact with expected difficulty to measure the geometrical component of the noise (I dont think its fair to compare the current noise measurement with direct current measurement like nonlinear hall effect.), I still think the manuscript is more suitable for publication in a more specialized journal after suitably revising the statement regarding the relationship with Ref. 38.

Authors' reply: As explained above, our shot noise formulation is different from Ref. 38. Particularly, our shift DSN features the intrinsic nature (the same as the shift current) and contains the geometric quantity—shift vector, showing a close relation with the shift current, while the DSN discussed in Ref. 38 does not contains these features.

In the above, we explain that the usual $S = 2eI$ noise for Ohmic current in metallic systems can not suppress the shot noise in insulating bulk systems discussed in our work. Furthermore, by taking a close look at the progress of shot noise measurements, we find that the **noninvasive** experimental technique recently used in Ref.[4] can be employed to measure the DSNs proposed in our work, where the possible $S = 2eI$ shot noises due to the metallic electrodes can be bypassed. Particularly, current fluctuations induced by nonequilibrium electrons (which in our setup are driven by the external optical field) generate fluctuating electromagnetic evanescent fields on the material surface, which can be detected by using a scattering type scanning near-field optical microscope called the scanning noise microscope⁴ without the introduction of metallic electrodes in the conventional noise experimental measurements. Combining all these facts, we hope that we addressed all the concerns raised by the reviewer.

In the revised manuscript, we have properly implemented the recent experimental technique in the discussion sections to experimentally confirm our proposal, see the third paragraph (highlighted in blue) in the section of [DISCUSSION].

Reply to the reviewer #3:

Reviewer's comments: Shot noise, a direct consequence of charge quantization, provides a unique lens to peer into quantum materials beyond what traditional conductance measurements offer. Specifically, shot noise experiments enable the determination of the charge and statistics of quasiparticles involved in transport, shedding light on the potential profile and internal energy scales within mesoscopic systems. This article discusses using light to detect the response of quantum materials, presenting a novel approach that involves the quantum fluctuation of photocurrent, specifically in the form of DC shot noise (SN), for the exploration of geometric properties in materials. The authors systematically develop a quantum theory and DFT calculations for DC shot noise, revealing and categorizing the resonant shift and injection DC shot noises. Additionally, they demonstrate that the expressions defining the shot noise susceptibility tensors are intricately linked to gauge-invariant geometric quantities. This is a well-written and timely article that makes a significant contribution to the field of quantum materials research. The authors' proposal to use DC shot noise as a probe for geometric properties is promising, and their theoretical framework is well-developed and convincing. This approach could be particularly valuable for materials that are difficult to probe using traditional methods. Overall, this is an excellent article that I recommend for publication. Undoubtedly, this work will captivate a diverse audience within the realm of quantum materials physics. Below, I list my comments and questions:

Authors' reply: We thank the reviewer for his/her careful reading and the recommendation on our manuscript. Below we will give a point-to-point reply for the comments and questions raised by the reviewer.

Reviewer's comments: 1. Firstly, reviewer #1 raised a pivotal query concerning the measurement of shot noise, prompting a need for a more detailed discussion by the authors on the setup of experimental verifications.

Authors' reply: We thank the reviewer for this suggestion. By taking a close look at the progress of shot noise measurements, we find that the recent experimental technique used in Ref.[4] may be employed to realize our proposal. In particular, current fluctuations induced by nonequilibrium electrons (which in our setup are driven by the external optical field) generate fluctuating electromagnetic evanescent fields on the material surface, which can be detected by using a scattering type scanning near-field optical microscope called the scanning noise microscope⁴ particularly without

the introduction of metallic electrodes in the conventional noise experimental measurements. In the revised manuscript, we have properly implemented the recent experimental technique in the discussion sections to experimentally confirm our proposal, see the third paragraph (highlighted in blue) in the section of [DISCUSSION].

Reviewer's comments: 2. In the context of an inversion-symmetric system with time reversal symmetry, where Berry curvature and shift vector vanish, the dominance of the quantum metric in shot noise becomes significant. Further exploration of this phenomenon, particularly in centrosymmetric systems where the 2nd order DC current response is absent, would be valuable for discussion.

Authors' reply: We thank the reviewer for this constructive suggestion. In systems with \mathcal{P} (inversion) and \mathcal{T} (time-reversal) symmetries, both the Berry curvature and the shift vector disappears due to $\mathcal{PT}\Omega_{mn}(\mathbf{k}) = -\Omega_{mn}(\mathbf{k})$ and $\mathcal{PT}\mathbf{R}_{mn}(\mathbf{k}) = -\mathbf{R}_{mn}(\mathbf{k})$, where Ω_{mn} and \mathbf{R}_{mn} stands for the Berry curvature and the shift vector, respectively. As a result, the dominant geometric quantity will be the quantum metric, as pointed out by the reviewer. Recently, the experimental observation for the intrinsic nonlinear Hall effect^{5,6} that driven by the quantum metric dipole has triggered much attention to explore the importance of quantum metric particularly in nonlinear responses. Note that the intrinsic nonlinear Hall effect can survive only in systems without \mathcal{P} -symmetry and \mathcal{T} -symmetry, while the quantum metric itself is not forbidden by these two symmetries. Therefore, the shot noise formulation developed in our work exactly offers a novel approach to probe the quantum metric in materials regardless of its \mathcal{P} and \mathcal{T} symmetries. In the revised manuscript, we properly implemented the above discussions, see the fourth paragraph (highlighted in blue) in the section of [DISCUSSION].

Reviewer's comments: 3. Regarding Figure 1 (d), the \mathbf{k} -resolved distribution exhibits positivity at $\hbar\omega=2.5\text{eV}$ throughout the Brillouin zone, but below 2eV , a negative value emerges in the mmm phase. The authors are encouraged to elucidate the underlying physics driving this behavior.

Authors' reply: We thank the reviewer for this important comment. The \mathbf{k} -resolved distribution in FIG.(1d) is plotting the integrand $\Delta_{nm}^a(\mathbf{k})(I_{mn}^{abc}(\mathbf{k}) + I_{mn}^{acb}(\mathbf{k}))\delta(\omega - \omega_{mn})$, where the resonant factor $\delta(\omega - \omega_{mn})$ implies that $\hbar\omega = \hbar\omega_{mn} = \epsilon_m - \epsilon_n$ and picks up the contributed \mathbf{k} points in the Brillouin zone by satisfying this relation. Note that the integrand can be positive or negative but FIG.(1d) seemingly exhibits the positive value throughout the Brillouin zone. This in fact

FIG. 1. The k -resolved distribution for the integrand of s^η with a photon energy $\hbar\omega = 2.5$ eV and $\hbar\omega = 1.44$ eV, respectively.

is due to the data visualization or plotting, where we have chosen a symmetric interval for the positive (dominant contribution) and negative (minor contribution) values and therefore the minor negative values has been suppressed. By breaking this artificial symmetry, we find that FIG.(d) indeed should contain a few sporadic negative values (near X point), as reproduced as FIG.(1a) of this reply. For a different incident light particularly with frequency $\hbar\omega = 1.44$ eV, the contributed k -distribution is changed, as shown in FIG.(1b) of this reply, where the k -solved distribution is mainly negative.

In the revised manuscript, we have replotted the FIG.(1d). In addition, FIG.(2e) and FIG.(2f) are replotted to avoid the same problem.

Reviewer's comments: 4. Notably, it appears that the authors did not incorporate spin-orbit coupling (SOC) in the DFT calculations. Given the substantial SOC in MoS₂, an inclusion of a discussion on the influence of SOC on shot noise would enhance the completeness of the study.

Authors' reply: We thank the reviewer for this constructive suggestions. In the revised manuscript, we have added the calculations for MoS₂ particularly by taking the spin-orbit coupling into account. And we find that the influence on MoS₂ is negligible, as can be seen from the FIG.(2) of this reply. In the revised Supplemental Material (see the last section), we have added this discussion.

FIG. 2. The (a) shift and (b) injection DSNs for MoS₂ without and with the consideration of spin-orbit coupling. Here ✓ indicates that the spin-orbit coupling is considered.

¹ Ya. M. Blanter and M. Büttiker, Phys. Rep. 1, 336 (2000).

² Q. Ma, A.G. Grushin, and K.S. Burch, Nat. Mater. 20, 16011614 (2021).

³ L.K. Shi, D. Zhang, K. Chang, and J. C. W. Song, Phys. Rev. Lett. 126, 197402 (2021).

⁴ Q.C. Weng *et al.*, Science 360, 775778 (2018).

⁵ A. Gao *et al.*, Science 381(6654): eadf1506 (2023).

⁶ N. Wang *et al.* Nature 621, 487492 (2023).

REVIEWERS' COMMENTS

Reviewer #3 (Remarks to the Author):

After careful consideration of the revisions made to the manuscript, I am pleased to note that the authors have addressed many of the concerns raised. The clarity and organization of the manuscript have notably improved, with enhanced explanations. Based on these improvements, I recommend the publication of the manuscript in NC.